# Network isolators inhibit failure spreading in complex networks

Franz Kaiser [1,2], Vito Latora [3,4,5,6] & Dirk Witthaut [1,2]✉

In our daily lives, we rely on the proper functioning of supply networks, from power grids to water transmission systems. A single failure in these critical infrastructures can lead to a complete collapse through a cascading failure mechanism. Counteracting strategies are thus heavily sought after. In this article, we introduce a general framework to analyse the spreading of failures in complex networks and demostrate that not only decreasing but also increasing the connectivity of the network can be an effective method to contain damages. We rigorously prove the existence of certain subgraphs, called network isolators, that can completely inhibit any failure spreading, and we show how to create such isolators in synthetic and real-world networks. The addition of selected links can thus prevent large scale outages as demonstrated for power transmission grids.

[1] Forschungszentrum Jülich, Institute for Energy and Climate Research (IEK-STE), Jülich, Germany. [2] Institute for Theoretical Physics, University of Cologne, Köln, Germany. [3] School of Mathematical Sciences, Queen Mary University of London, London, UK. [4] Dipartimento di Fisica ed Astronomia, Università di Catania and INFN, Catania, Italy. [5] The Alan Turing Institute, The British Library, London, UK. [6] Complexity Science Hub Vienna, Vienna, Austria. ✉email: d.witthaut@fz-juelich.de

Complex networked systems are subject to external perturbations, damages or attacks with potentially catastrophic consequences[1,2]. The loss of even a single edge can cause a blackout in a power grid[3,4], the dieback of a biological network[5], or the collapse of an entire ecological network[6]. It is thus essential to understand how the structure of a network determines its response to perturbations and its global resilience[7–11]. Here, we propose a general framework to model the redistribution of flows in a complex network that follows a small and local failure, and we suggest novel and more efficient strategies to improve network resilience. Our findings reveal that propagation of damages can be better limited by adding selected links instead of removing links and can turn very useful to construct more robust networks or to improve existing ones.

The division of a network into weakly coupled parts provides the most intuitive method to inhibit the spreading of failures, thus improving system resilience[12–15]. An example is shown in Fig. 1a for an elementary supply network with two weakly connected modules. The response to an edge failure is strong locally, but it is reduced in the other module of the network which has only few links connecting to the part where the failure happened. A similar effect is observed in a real-world case: the Scandinavian power grid in Fig. 1d. The study of community structures in both natural and man-made systems is an integral part of network science: a variety of methods has been developed to define and identify the weakly connected modules of a network[16–18], and the important role of community structures in network dynamics is today well recognised.

Limiting connectivity for the sake of additional security is, however, not always desirable. For instance, microgrid concepts and intentional islanding are heavily discussed in energy systems research[19,20], but the overall demand for electric power transmission actually increases[21,22]. Other methods to contain perturbations or damages in complex networks are thus needed. Indeed, an exceptionally strong interconnectivity between two modules can also suppress failure spreading as shown in Fig. 1b, e. Notably, a strong interconnectivity can be realised in different ways. In the random network example in Fig. 1b, a high number of links connects a subset of nodes of the two modules. In real

vascular networks of leaves the suppression of failure spreading occurs naturally because the central vein between the left and right parts has an exceptionally large weight (Fig. 1e, cf. also[23]).

Remarkably, failure spreading can be completely stopped by certain subgraphs which we refer to as *network isolators* in the following, an example being shown in Fig. 1c. The failure of an edge in the right part of the network does not affect the flows in the left part at all. Real world networks can be made perfectly resistant to failure propagation by the ad-hoc addition of few links to create network isolators, as demonstrated for the Scandinavian power grid in Fig. 1f consisting of three weakly coupled modules. The suppression of failure spreading is readily generalised to networks with more than two modules.

## Results

**A model for supply networks.** Our results are based on a general framework that allows a theoretical analysis of the interplay of network connectivity and robustness in the context of supply or transportation networks. Consider a simple graph $G$ with edge set $E$ and vertex set $V$ consisting of $L = |E|$ edges and $N = |V|$ vertices. Many such systems can in fact be modelled by linear flow networks where the flow over an edge $e = (i, j) \in E(G)$ depends linearly on the gradient of a potential function across the edge,

$$F_{i \to j} = A_{ij} \cdot (\vartheta_i - \vartheta_j). \tag{1}$$

In particular, this description applies to power transmission grids[2,24–26], where $F$ is the real power flow, $\vartheta_i$ denotes the nodal voltage phase angle and $A_{ij}$ is given by the line susceptance. Nonlinear effects in electric power grids will be discussed below. Furthermore, the description (1) applies to hydraulic and vascular networks[27,28], where $F$ is the flow of water or nutrients, $\vartheta_i$ is the local pressure and $A_{ij}$ the edge's weight. Equivalent problems arise in the linearisation of general diffusively coupled networks of dynamical systems around an equilibrium or limit cycle[29]. We discuss these and other applications of linear flow models in detail in Supplementary Note 1.

Now assume that there are sources and sinks attached to the nodes in the network $P_i \in \mathbb{R}$, $i \in V(G)$ where $P_i > 0$ represents a

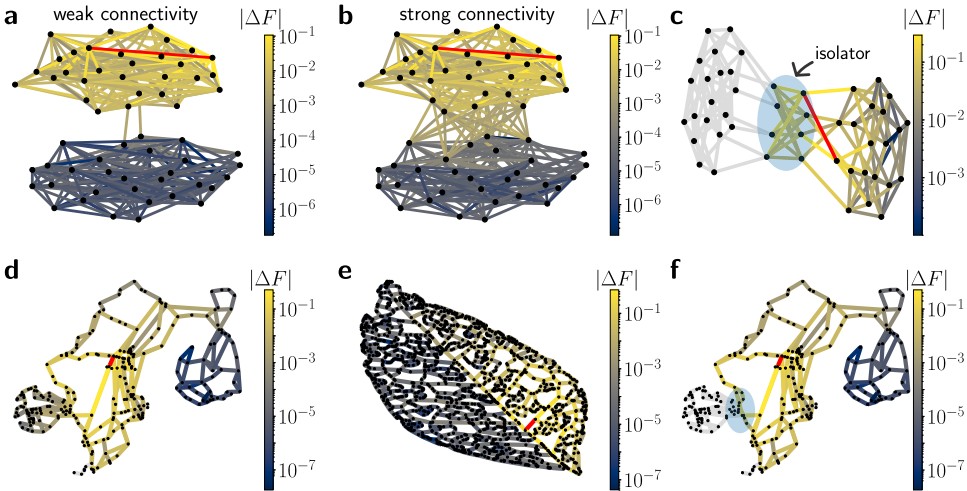

**Fig. 1 Different network structures inhibit the spreading of failures in complex networks.** We simulate the impact of a single failing link (red) for different network structures; resulting flow changes are colour coded. **a**, **b** Both a weak and a strong interconnectivity can suppress the spreading of failures between two modules of a complex network. **c** Failure spreading is prevented completely by a network isolator (blue shading); flow changes on the grey links are exactly zero. **d** The Scandinavian power grid consists of three weakly connected modules, which suppresses failure spreading between the modules[44]. **e** The vascular network of a *Bursera hollickii* leaf contains a strong central vein[47], which suppresses failure spreading between the two sides of the leaf. **f** Same as in (**d**) but with the addition of two links (blue shading) to create a network isolator. See Methods and caption of Fig. 2 for further information on the graphs used here.

source and $P_i < 0$ a sink. Then the flows at each node have to balance with the sources and sinks

$$P_i = \sum_{j=1}^{N} F_{i \to j} \qquad \forall i \in V(G). \tag{2}$$

This equation is known as continuity equation or *Kirchhoff's current law*. If the sources and sinks $P_i$ are given, Eqs. (2) and (1) completely determine the potentials in the network (up to a constant shift to all potentials). In a power grid, the sources and sinks are the power injections or withdrawals as a result of power production or consumption, respectively. When looking at the stable, operational fixed point of a power grid they are balanced such that

$$\sum_{i=1}^{N} P_i = 0, \tag{3}$$

we therefore assume this to hold in the following sections.

For further use, we introduce a compact vectorised notation, defining the vector of injections $\overrightarrow{P} = (P_1, \ldots, P_N)^\top$ and the vector of potentials $\overrightarrow{\vartheta} = (\vartheta_1, \ldots, \vartheta_N)^\top$, where the superscript $\top$ denotes the transpose. The coupling coefficients $A_{ij} = A_{ji}$ are summarised in the weighted adjacency matrix $\boldsymbol{A} \in \mathbb{R}^{N \times N}$. Furthermore, we define the diagonal matrix $\boldsymbol{D} \in \mathbb{R}^{N \times N}$ with entries $D_{ii} = \sum_j A_{ij}$ as well as the weighted graph Laplacian[30]

$$\boldsymbol{L} = \boldsymbol{D} - \boldsymbol{A}. \tag{4}$$

Kirchhoff's equations then assume the compact form

$$\boldsymbol{L} \overrightarrow{\vartheta} = \overrightarrow{P}. \tag{5}$$

Notably, the Laplacian matrix is also useful to infer the large scale connectivity and the community structure of a given network[31].

**Modelling link failures**. The impact of a damage in linear flow networks can be calculated analytically. Assume that an edge $\ell = (r, s)$ fails, and summarise the response at all nodes $i = 1, \ldots, N$ in terms of the vector of changes in nodal potentials $\Delta \overrightarrow{\vartheta} = (\Delta\vartheta_1, \ldots, \Delta\vartheta_N)^\top$. The response can be calculated by subtracting Eq. (5) for the new and the old network which yields after some manipulations (Ref. [25], Supplementary Note 2)

$$\boldsymbol{L} \Delta \overrightarrow{\vartheta} = q_\ell \overrightarrow{\nu}_\ell, \tag{6}$$

where $\overrightarrow{\nu}_\ell$ is a vector with $+1$ at position $r$ and $-1$ at position $s$, and $q_\ell = 1 - A_{rs} \overrightarrow{\nu}_\ell^\top \boldsymbol{L}^{-1} \overrightarrow{\nu}_\ell$ is a source strength[25]. We thus find that the response of a network to failures is essentially determined by the Laplacian $\boldsymbol{L}$.

To quantify the effect of connectivity on failure spreading, we have studied the impact of different failures in a variety of synthetic networks as well as in several real-world networks. For a given initial failure of an edge $\ell$, we compute the flow changes

$$\Delta F_{i \to j} = A_{ij} \cdot (\Delta\vartheta_i - \Delta\vartheta_j) \tag{7}$$

for all edges $e = (i, j)$ in a given subgraph $G'$ of the network. Furthermore, we must take into account that the impact of a failure generally decreases with distance[25,32,33]. As an overall measure of the impact of a failure we thus consider the expression $\langle |\Delta F_{i \to j}| \rangle_d^{(i,j) \in G'}$, which gives the magnitude of flow changes averaged over all edges $(i, j) \in G'$ at a given distance $d$ to the edge $\ell$ (see Methods for details on the notion of distance used here). The prime question is now whether the impact differs substantially between the communities or moduli of a network. Here, we assume that the moduli or communities are known for the network under consideration and thus do not address the

question how to determine them. We thus plot the ratio

$$R(\ell, d) = \frac{\langle |\Delta F_{i \to j}| \rangle_d^{(i,j) \in \mathrm{O}}}{\langle |\Delta F_{i \to j}| \rangle_d^{(i,j) \in \mathrm{S}}}. \tag{8}$$

between the module of the network $G' = \mathrm{O}$ without initial failures and the module $G' = \mathrm{S}$ containing the failing edge $\ell$. If this ratio approaches or reaches zero, this is indicative of a very strong suppression of failure spreading into the other part of the network.

**The impact of network connectivity on failure spreading**. To study how the impact of failure spreading depends on the network structure, we considered synthetic graphs obtained by connecting two Erdős–Rényi (ER) random graphs to each other at preselected, randomly chosen vertices with a tunable probability $\mu \in [0, 1]$[34] (see Methods). The resulting graphs have a connectivity structure ranging from two weakly connected communities for low values of $\mu$ shown in Fig. 1a to strongly connected parts shown in panel b. In the limit $\mu = 1$, the two modules are connected via a complete bipartite graph as shown in Fig. 1c. This is a possible realisation of a *network isolator*, since it completely suppresses flow changes. We will explain the concept of *network isolators* and provide a rigorous definition in the next section.

The corresponding adjacency matrices clearly indicate the connectivity structure, revealing the strong or weak coupling between the two modules of the networks (Fig. 2a, b, d). Remarkably, evaluating the quantity $R(\ell)$, obtained by averaging the ratio over flow changes $R(\ell, d)$ over all distances $d$ for a specific trigger link $\ell$, for a varying connectivity structure tuned by $\mu$, we find that the spreading of failures is largely suppressed for both weak and strong connectivity between the two modules as shown in Fig. 2c. Note that this finding is not limited to the interconnectivity of two modules, but can be readily generalised to three—or more—modules as we demonstrate in Supplementary Fig. 3. Distance plays a minor role for the ratio of flow changes $R(\ell, d)$ as illustrated in Supplementary Fig. 2.

**Network isolators inhibit failure spreading**. Network symmetries are known to play an important role for the dynamics and synchronisability of a network[35–37]. Network isolators as a specific connectivity structure completely inhibit the spreading of failures from one network module to another. They manifest also as particular, symmetric patterns in the region of the adjacency matrix describing the connectivity between the two parts of the network as we have seen in Fig. 2d. To see this, we make use of Eq. (4) to rewrite the Laplacian matrix $\boldsymbol{L}$ of the entire network as follows

$$\boldsymbol{L} = \begin{pmatrix} \boldsymbol{L}_1 + \boldsymbol{D}_1 & -\boldsymbol{A}_{12} \\ -\boldsymbol{A}_{12}^\top & \boldsymbol{L}_2 + \boldsymbol{D}_2 \end{pmatrix} \tag{9}$$

Here, $\boldsymbol{L}_1$ and $\boldsymbol{L}_2$ are the Laplacian matrices of the two parts of the network which consist of $N_1$ and $N_2$ nodes, $\boldsymbol{A}_{12} \in \mathbb{R}^{N_1 \times N_2}$ is the region of the weighted adjacency matrix encoding the connectivity between the two parts of the network and $\boldsymbol{D}_1$ and $\boldsymbol{D}_2$ are the degree matrices of these mutual connections, i.e. the matrices containing the nodes' weighted degrees on the diagonals. Then network isolators are characterised by the following theorem.

*Theorem 1* Consider a linear flow network composed of two modules 1,2 and let $\boldsymbol{A}_{12}$ denote the weighted adjacency matrix of the mutual connections as described in Eq. (9). An edge failure in one module does not affect the flows in the other module if rank $(\boldsymbol{A}_{12}) = 1$. For unweighted networks this criterion is fulfilled if

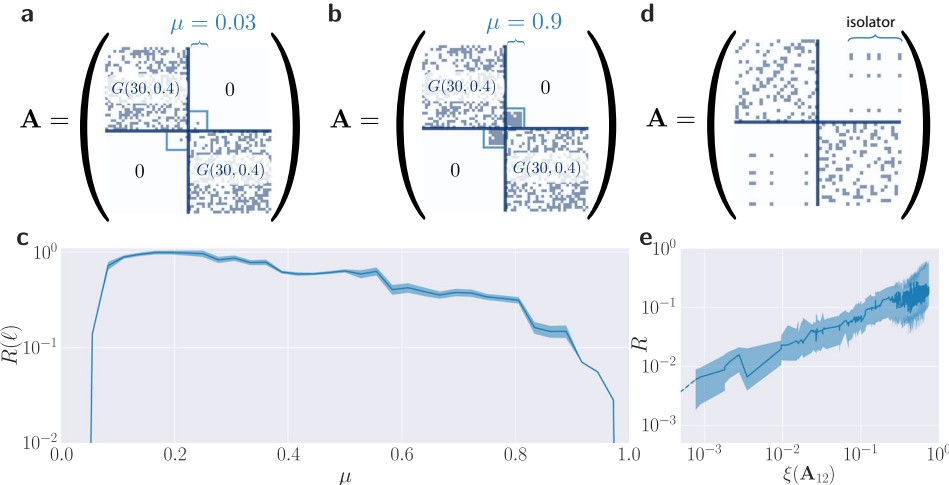

**Fig. 2 Effectiveness and robustness of shielding network structures. a, b** Adjacency matrices for the graphs shown in Fig. 1a, b. Two random graphs $G(30,0.4)$ are inter-connected via a fraction $c = 0.2$ of their nodes chosen at random, and links are added with probability $\mu$, interpolating between weak (**a**) or strong (**b**) interconnectivity (see Methods for details). **c** The average ratio of flow changes $R(\ell)$ in the two components (Eq. (8)) is strongly suppressed for both high and low interconnectivity $\mu$. The blue line represents the median value over all distances and the shaded region indicates the 0.25- and 0.75-quantiles. **d** Adjacency matrix for the six-regular graph shown in Fig. 1c and containing a network isolator. Note that all nodes in the graph including those in the network isolator have degree equal to six, which allows us to exclude any potential impact of heterogeneity in the degree on failure spreading in this case. **e** The ratio of flow changes $R$, now averaged over all possible trigger links $\ell$ and distances $d$, vanishes for a network isolator described by the condition $\xi(\mathbf{A}_{12}) = 0$ and increases algebraically with the coherence parameter $\xi$ (cf. Eq. (10)) when perturbed (see Methods for details on the simulation). Again, median and 0.25- and 0.75-quantiles are shown resulting from averaging over all distances and then trigger links.

$\mathbf{A}_{12}$ describes a complete bipartite graph. The subgraph connecting the two modules is referred to as a *network isolator*.

A proof can be found in Supplementary Note 3. Note that, while network isolators prevent failure spreading, we found that they do not influence network controllability as we illustrate in Supplementary Note 4 and Supplementary Fig. 8.

Since most real world examples of networks do not contain perfect network isolators, we have studied the robustness of a network isolator against modifications of the topology. Starting from a unit rank matrix, we perturb the adjacency matrix $\mathbf{A}_{12}$ iteratively (see Methods for details). The deviation of the perturbed matrix $\mathbf{A}_{12}$ from a unit rank matrix is then quantified using its coherence statistics defined as[38],

$$\xi(\mathbf{A}_{12}) = 1 - \min_{i,j} \frac{\langle \vec{a}_i, \vec{a}_j \rangle}{\| \vec{a}_i \| \| \vec{a}_j \|}, \quad (10)$$

where $\vec{a}_i, i = 1, \ldots, m$ are the matrix columns. Note that the latter expression $\cos(\angle \vec{a}_i, \vec{a}_j) = \frac{\langle \vec{a}_i, \vec{a}_j \rangle}{\| \vec{a}_i \| \| \vec{a}_j \|}$ may also be interpreted via the angle between two matrix columns, $\vec{a}_i$ and thus $\xi(\mathbf{A}_{12})$ approaches a value of unity if all columns are parallel. The performance of the isolator is then measured by calculating the ratio of flow changes $R$, which is obtained from $R(\ell, d)$ by averaging over all possible trigger links and distances. A perfect isolator is characterised by $\xi(\mathbf{A}_{12}) = 0$ and enables a complete containment of failure spreading such that $R = 0$. For perturbed isolators, we find that $R$ increases approximately algebraically with $\xi(\mathbf{A}_{12})$, see Fig. 2e and Supplementary Fig. 5. Hence, the isolation effect persists for small perturbations, albeit with reduced efficiency. Note also that network isolators are not limited to two connected modules, but can be readily generalised to the interconnectivity of three—or more—modules that are mutually shielded against failures as we demonstrate in Supplementary Fig. 4. Finally, we illustrate that network isolators do not increase the vulnerability of a network in case a link located in the isolator fails in Supplementary Fig. 6.

**Constructing network isolators in real-world graphs.** Network isolators are not limited to the particular situation shown in Fig. 1. In Fig. 3a–c, we identify several subgraphs that allow to easily introduce network isolators into existing topologies. For subgraphs with a prior low connectivity, as measured by a small vertex cut (Fig. 3a) or a small edge cut (Fig. 3b, c), network isolators may be introduced with small network modifications— by adding (a,b) or removing and adding (c) selected links with weights adjusted such that Theorem 1 is fulfilled. For a given graph these recipes may thus be applied as follows: (1) Identify modules of the graph that are weakly connected to one another as measured by a low vertex cut or edge cut of the vertices or edges connecting them. (2) Depending on the target—e.g. whether building new edges or vertices is costly or, on the other hand, a minimum connectivity between the modules is required after the modification—identify the optimal strategy to achieve a complete bipartite connectivity between the modules by adding or removing vertices and edges. Here, the recipes shown in the Figure may be applied directly if the prior connectivity has the indicated edge or vertex cuts. (3) Tune the edge weights such that $\text{rank}(\mathbf{A}_{12}) = 1$ is achieved, i.e. a network isolator is realised.

We illustrate each of the strategies in real-world power grids. We consider the British grid (d), the Scandinavian power grid (e) and the Central European power grid (f) and add a network isolator to each of the networks by making use of the strategies shown in panels a–c. We then simulate the failure of a single link to illustrate that network isolators suppress failure spreading in each situation. Thus, network isolators can be used to make various real-world power grids more resilient to failures. In a Supplementary Fig. 7, we compare the situation with the isolator to the situation before constructing the isolator for each of the networks.

**Network isolators suppress cascade propagation.** Perfect network isolators can be easily constructed to improve the resilience of complex networked systems. As a practical example we show an application to electric power grids, where large scale blackouts

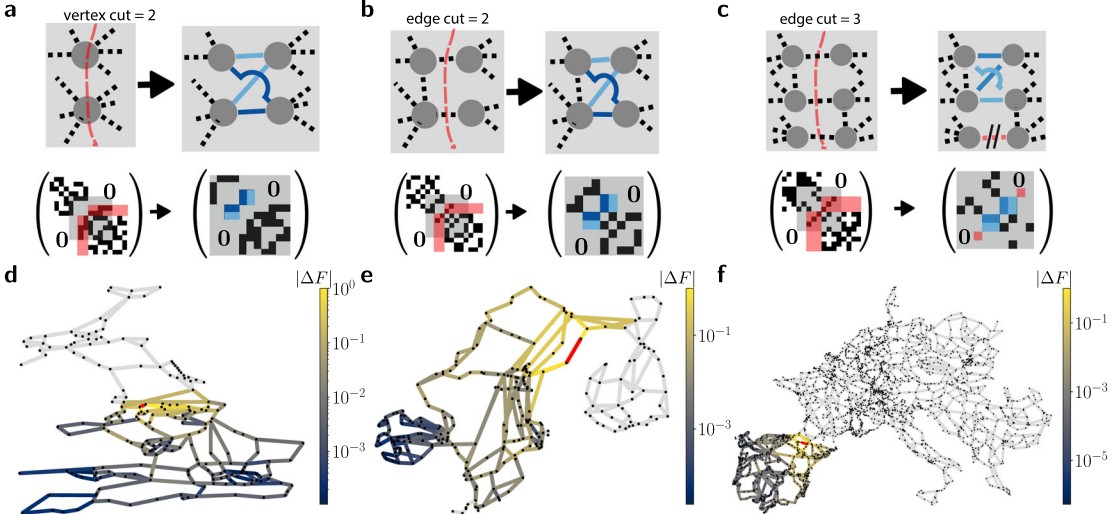

**Fig. 3 Different ways of constructing isolators in real-world power grids. a–c** Alternative methods of creating an isolator in a given network. We show the network structure before (top left) and after (top right) the addition of a network isolator, as well as corresponding adjacency matrices (bottom) with the different shades of blue representing the weight $A_{ij}$ of the respective edge. A lower prior connectivity simplifies the creation of isolators as measured by the vertex cut (**a**) or edge cut (**b, c**) which is visible in the adjacency matrix (entries colored red). The creation of network isolators results in characteristic patterns in the adjacency matrix in terms of the capacities of the isolator edges (shades of blue). **d–f** Realisation of network isolators in real-world power grids. We construct network isolators in the British power grid (**d**), the Scandinavian power grid (**e**) and the Central European power grid (**f**) using the recipes illustrated in (**a–c**). For each power grid, we colour code the flow changes after the failure of a single link carrying a unit flow (red). In each case, the network isolator inhibits flow changes, i.e. $\Delta F = 0$, (light grey) in the part of the network that is shielded by the isolator.

are typically triggered by the outage of a single transmission element which leads to a cascade of failures[3,39]. We demonstrate the impact of network isolators against cascading failures in the case of the Scandinavian grid.

In the original grid layout, the modules are weakly connected, thus failure spreading between these modules is reduced—but it is possible. A failure in one area can spread to other areas and cause a global cascade of failures, as demonstrated in Fig. 4a, b for a cascade emerging in Western Norway. This spreading may in principle be prevented by decoupling different areas of the grid, but this is highly undesired. In fact, future energy systems will require more connectivity, not less, to transmit an increasing amount of renewable electric power[21,22]. In contrast, building a network isolator can completely inhibit failure spreading at increased connectivity. A perfect isolator can be realised with moderate effort by reconstructing two substations in Norway, such that they effectively form two nodes each. The new nodes must be linked by internal connections and one additional two-circuit overhead line, whose parameters are optimised such that the condition $\text{rank}(\mathbf{A}_{12}) = 1$ is satisfied (Fig. 4c). A simulation for such an optimised grid layout shows that the spreading of the cascade is completely suppressed (Fig. 4d). The network remains connected and load shedding is no longer necessary as a containment strategy[2,3]. To demonstrate that network isolators effectively suppress cascade propagation for different networks and initial failure patterns, we evaluate the statistics of cascade sizes in networks with and without network isolators (see Supplementary Fig. 9). To analyse how the relatively localised flow changes involved here lead to a non-local cascade, individual steps of the cascade are shown in Supplementary Fig. 12.

**Network isolators beyond linear flow networks.** The concept of network isolators has been established for linear flow networks, but can be extended in two ways. (1) We can rigorously prove that network isolators determine the response to structural damages for a class of non-linear networked dynamical systems with diffusive coupling. More precisely, the isolator effect is still rigorously

valid if the dynamics of a node $i$ depends on the state of the other nodes $x_j$ only through the term $f_i(\sum_j L_{ij} x_j)$, where $L$ is the Laplacian and the function $f_i$ satisfies $f_i(0) = 0$, but is arbitrary otherwise (see Supplementary Note 3, Corollary 2). (2) For many non-linear systems of practical importance, the impact of failures or perturbations is well described by a linearisation around an equilibrium or limit cycle (see ref. [29]) for which an approximate isolation can be achieved (see Supplementary Note 3, subsection 4).

To systematically analyse how non-linearity affects failure spreading through network isolators we first consider a natural extension of the linear flows in Eq. (1), replacing the linear coupling by its sinusoidal counterpart

$$\tilde{F}_{i \to j} = A_{ij} \cdot \sin(\vartheta_i - \vartheta_j), \tag{11}$$

which yields the well-known Kuramoto model[40,41]. If phase differences between neighbouring vertices are small, one can expand the sine function as $\sin(\vartheta_i - \vartheta_j) = (\vartheta_i - \vartheta_j) + \mathcal{O}((\vartheta_i - \vartheta_j)^3)$ (see Supplementary Note 1). Hence, our previous result remain valid to linear order, whereas a higher degree of non-linearity may gradually weaken the effects. In particular, the effectiveness of a network isolator depends on the relative load of the edges $|\tilde{F}_{i \to j}|/A_{ij}$. We study this numerically by increasing the injections $P_i$ at all nodes proportionally, thus increasing the relative edge loads and the importance of the non-linearity of the sine function.

We then analyse the non-linear flow changes $\Delta \tilde{F}(\ell)$ after the failure of a single link for different degrees of non-linearity in Fig. 5a, b (light to dark lines). To systematically evaluate the degree of non-linearity, we analyse the maximal absolute non-linear flow $|\tilde{F}|_{\text{max}}$ in the entire network. Due to the sinusoidal character of the coupling (see Eq. (11)) and since edge weights are set to unity for the Figure, a relative loading close to unity indicates a highly non-linear system. As expected, the flow changes decrease with distance independently of the non-linearity. However, even for the strongest degree of non-linearity considered here, flow changes in the module shielded by the isolator are still

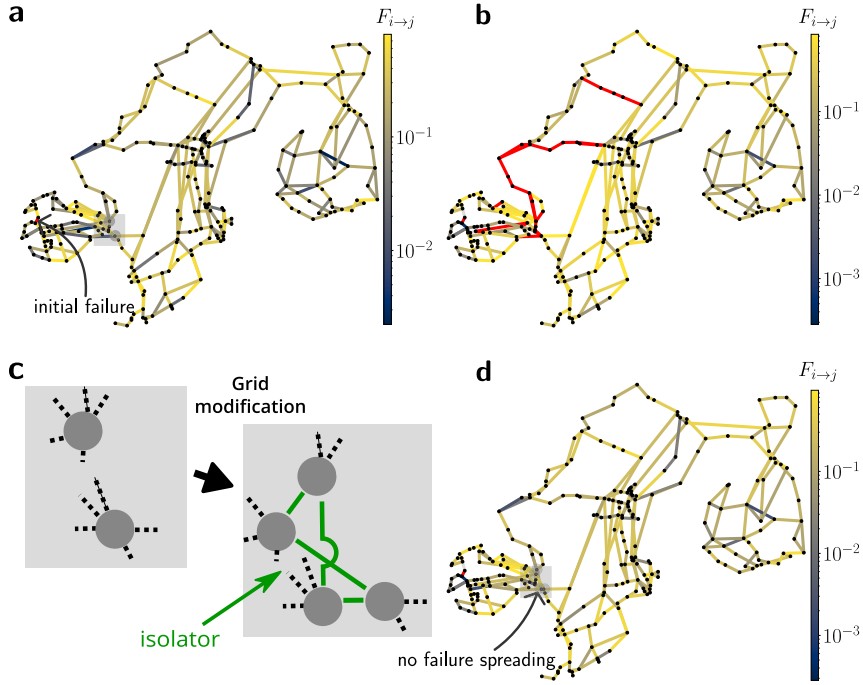

**Fig. 4 Network isolators can contain cascading failures in power grids. a** Line loading (colour code) on the Scandinavian grid in units relative to maximal loading before the initial failure of a single line (coloured red). **b** The initial failure results in a cascade of overloads (red coloured lines) until the grid disconnects into several parts. **c** Magnification of the grid structure in Eastern Norway (grey box, **a**). A small modification of the grid enables the construction of a network isolator: adopting the recipe presented in Fig. 3a, we select two nodes (left) that are further split up into two separate nodes each which are mutually connected via a network isolator by adding four edges (right, green). Note that the removal of these two nodes would disconnect the network into two separate parts, i.e. they form a vertex cut of size two. **d** Introducing the network isolator completely suppresses the spreading of failures from Western Norway to the rest of the grid thus inhibiting the cascade observed in (**b**). The first two steps of this cascade are shown in Supplementary Fig. 12.

several orders of magnitude lower than at the same distance in the module containing the trigger link. We confirm this result by evaluating the non-linear version of the flow ratio (8) for different graphs, network conditions and degrees of non-linearity in Supplementary Fig. 10. Furthermore, we demonstrate that introducing a network isolator may slightly improve the system's resilience against dynamically induced failures due to transient overloads in Supplementary Fig. 11.

We now study the robustness of this effect in several regards and elucidate possible ways to designing robust network isolators for non-linear systems. The condition $\text{rank}(\mathbf{A}_{12}) = 1$ allows for different possible realisations of network isolators in terms of the edge weights. In linear flow networks, all these realisations are equally efficient: They completely suppress flow changes in the module shielded by the network isolator by virtue of Theorem 1. But which combination of edge weights provides the strongest isolating effect in weakly non-linear systems?

To examine this question systematically we consider a simple but non-trivial realisation of a network isolator where two nodes in one module are connected to two nodes in the other module (see e.g. Fig. 3a, right). The isolator is thus formed by four edges, and we fix the overall possible available edge weight to build the network isolator to a constant value $\mathfrak{A} \in \mathbb{R}$. Hence, the weights of the four edges in the isolator have to satisfy two conditions,

$$a_1 + a_2 + a_3 + a_4 = \mathfrak{A} \quad \text{and} \quad a_1 a_4 - a_2 a_3 = 0,$$

leaving two degrees of freedom to optimise the isolator performance (see Methods for details). In Fig. 5c we examine the network isolator's performance measured by the averaged, non-linear flow changes in the module shielded by the isolator for all possible failing links in the other module for a weakly non-

linear system with flows described by Eq. (11). On the other hand, we analyse the worst-case available $N-1$ weight, i.e. the overall edge weight connecting the two modules if the edge in the network isolator with the largest weight fails. We find that network isolators with strongly heterogeneous edge weights $a_1$ and $a_2$ inhibit failure spreading the most in the weakly non-linear system under consideration. However, the uniform choice $a_i = \mathfrak{A}/4, i \in \{1, 2, 3, 4\}$ yields the highest the available $N-1$ weight, while still inhibiting failure spreading relatively strongly. Note that other choices to estimate the impact of removing a single link in the network isolator, e.g. the size of the cascade caused by the failure of the link in the isolator or the reduction in shielding provided by the isolator after the failure might come to a different conclusion which choice of weights yields the "best" network isolator.

We now further extend the results on non-linear systems by considering the full load flow equations that describe power flows in power grids with line losses. The results of the numerical simulations are reported in Fig. 6: First, we consider the impact of a single failing line for a realistic dispatch and realistic line weights in the British power grid without any modification, where flows are now evaluated based on the full non-linear AC load flow[42]. For a given vertex $i \in V(G)$ they are calculated as (Supplementary Note 1, Eq.(8))

$$P_i = \sum_{k=1}^{N} |V_i||V_k|(G_{ik}\cos(\vartheta_i - \vartheta_k) + B_{ik}\sin(\vartheta_i - \vartheta_k)),$$

$$Q_i = \sum_{k=1}^{N} |V_i||V_k|(G_{ik}\sin(\vartheta_i - \vartheta_k) - B_{ik}\cos(\vartheta_i - \vartheta_k)). \tag{12}$$

Note that this set of equations again reduces to the linear flow

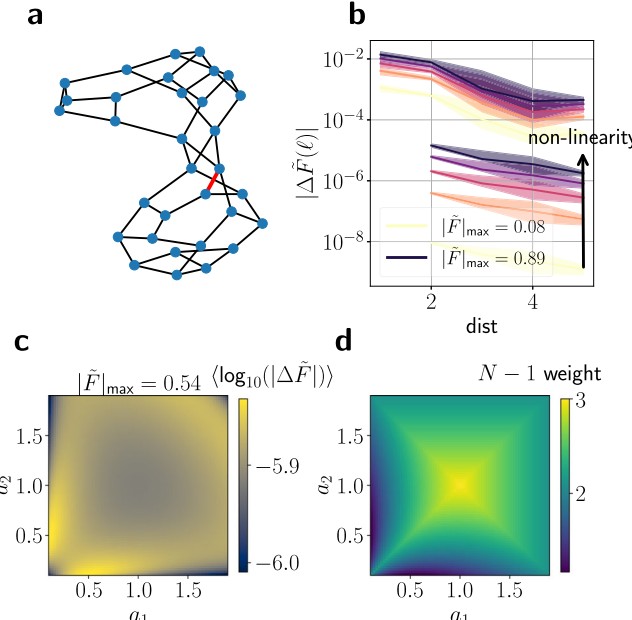

**Fig. 5 Robust design of network isolators in the Kuramoto model. a** To study the effect of non-linearity on network isolators, we simulate the failure of a single link (red) in a network consisting of two modules that are connected via a network isolator. **b** We consider the median absolute non-linear flow changes $|\Delta \tilde{F}(\ell)|$ (Eq. (11)) on a link $\ell$ after the removal of the link shown in (**a**). We analyse the effect of edge distance to the failing link (x-axis) and increasing degree of non-linearity (colour code from light to dark). We compare the flow changes in the lower module that contains the failing link (curves on the upper left) and the isolated module (curves on the lower right) by averaging the flow changes over all links in the given module at a fixed distance. As expected, flow changes in the upper module are lowest for a weakly non-linear system (bright line) and increase with the non-linearity, but a strong isolation effect persists even for a high degree of non-linearity (dark purple line). Shaded region indicates the 0.25- and 0.75-quantiles evaluated over the given distance. **c** We fix the overall available edge weight of the four edges forming the isolator to $\sum_i a_i = 4$ and systematically scan over the remaining degrees of freedom, measuring the isolator performance in terms of the mean logarithmic flow changes $\langle \log_{10}(|\Delta \tilde{F}|) \rangle$ for a fixed degree of (intermediate) non-linearity. We observe that a heterogeneous isolator where the weights differ strongly provides the best shielding. **d** We evaluate the available worst-case $N-1$ weight, i.e. the overall edge weight connecting the two modules after the failure of a single link in the isolator, for the same set of edge weights as in (**c**). Here, isolators with homogeneous weights perform best. Edge weights of all non-isolator edges are set to unity, $A_{ij} = 1, \forall (i,j) \in E(G)$ in all panels.

model in Eqs. (2) and (1) in the so-called DC approximation (see Supplementary Note 1). As a result, failures spread to both the Northern part of the power grid and the Southern part equally (panel a). After introducing a network isolator by adding two links, flow changes are completely suppressed in the linear approximation of power flows (panel b), but also significantly reduced when calculating the flow changes based on the full non-linear AC power flow: Comparing the non-linearly calculated flow changes in the initial scenario and the scenario with the isolator, we observe an ~100-fold reduction at all distances to the failing link in the module shielded by the network isolator (panel d). Thus we conclude that isolators also suppress failure spreading in non-linear models.

## Discussion

In conclusion, connectivity determines the resilience of complex networks in manifold ways. As expected, a division of a network into weakly coupled modules suppresses the spreading of failures from one module to the others. Remarkably, we have found that a strong interconnectivity can equally well suppress the spreading in both flow networks and in networks of non-linear dynamical systems. We have demonstrated that an even stronger effect can be obtained by certain subgraphs called isolators, which completely inhibit the spreading of failures in linear systems.

We then showed that isolators can be easily created in a network to mitigate cascading failures, for instance in electric power grids, while enabling an arbitrary degree of connectivity between the different parts of the network. These results widen our perspective on the large scale organisation of complex networks in general, showing that very diverse structural patterns can exist that isolate functional units and improve network resilience.

Furthermore, our results show that algebraic properties of networks can have striking effects on their function and robustness—depending on the type of flow model. Similar effects are not present in simple models where flows are rerouted along the shortest paths only[4,9], but they can become essential in physical supply network models where various paths contribute and interact in a non-trivial way.

## Methods

**Creating graphs with strong or weak inter-module connectivity**. We introduce a model to create ensembles of graphs consisting of two subgraphs with weak or strong interconnectvity similar to the approach in ref. [43], see Figs. 1 and 2. We start with two disconnected Erdős–Rényi random graphs $G_1(N_1, p_1)$ and $G_2(N_2, p_2)$, where $N$ denotes the number of nodes in the graph and $p$ the probability that two randomly chosen nodes are connected by an edge[34]. Then we randomly choose $n_1 = [c \cdot N_1]$ nodes $v = \{v_1, ..., v_{n_1}\}$ in $G_1$ and $n_2 = [c \cdot N_2]$ nodes $w = \{w_1, ..., w_{n_2}\}$ in $G_2$. Here, $c \in [0,1] \subset \mathbb{R}$ is a constant representing the share of nodes connecting

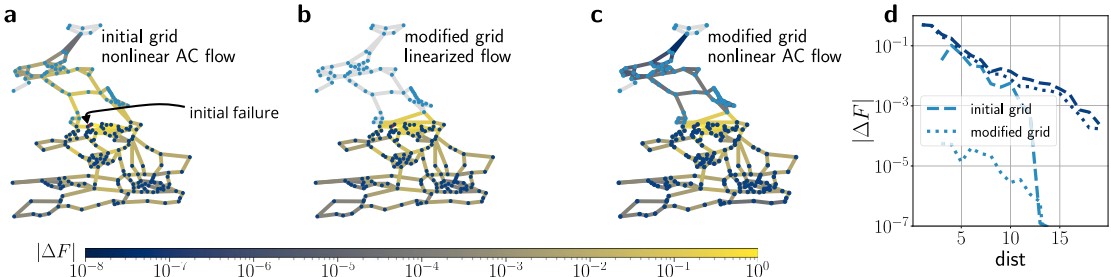

**Fig. 6 Network isolators suppress failure spreading in full non-linear AC load flow. a** An initially failing link with unit flow (red) in the British power grid results in changes of real power flow (colour code) throughout the whole network, as obtained by computing a non-linear full AC power flow[44]. **b, c** After introducing a network isolator based on the strategy presented in panel (**a** of Fig. 3), failure spreading is perfectly inhibited in the linear power flow approximation, and still significantly reduced in the non-linear full AC load flow. **d** We compare the median absolute flow changes, calculated using the non-linear load flow (Eq. (12)), after the failure of the link in the initial grid (dashed lines, **a**) and the modified grid (dotted lines, **c**). Whereas the flow changes in the lower module of the power grid (dark blue nodes) stay approximately the same after the grid modification (dark blue lines), they are significantly reduced in the grid's upper module (light blue nodes) that is shielded by the network isolator (light blue lines).

to the other subgraph and $[\cdot]$ denotes the nearest integer. Out of all possible edges $e = \{(v_1, w_1), \ldots, (v_{n_1}, w_1), \ldots, (v_{n_1}, w_{n_2})\}$ between the two sets of nodes $v$ and $w$, we randomly add a share of $\mu \in [0, 1]$. The parameter $\mu$ controls the connectivity of the two subgraphs $G_1$ and $G_2$: They remain disconnected for $\mu = 0$ and they are connected via a complete bipartite graph for for $\mu = 1$. For $c = 1$ and $\mu = p_1 = p_2$ we recover a single Erdős–Rényi random graph with $N = N_1 + N_2$ nodes. Note that this procedure is in principal not limited to ER random graphs. We apply it to study other types of graphs as shown in Supplementary Fig. 1.

**Calculating the distance between edges.** The notion of distance used throughout the manuscript is the unweighted edge distance. This notion of distance measures the length of the shortest path between two edges $\ell = (r, s)$ and $e = (m, n)$ and is defined as follows[25]

$$\text{dist}(\ell, e) = \min_{v_1 \in \{r,s\}, v_2 \in \{m,n\}} \text{d}(v_1, v_2) + 1, \tag{13}$$

where $\text{d}(v_1, v_2)$ is the unweighted shortest-path or geodesic distance between nodes $v_1$ and $v_2$ and the addition of unity ensures that neighbouring edge have a non-vanishing distance.

**Perturbing network isolators.** The robustness of network isolators to structural perturbations is analysed as follows. Let $G = (E, V)$ be a graph whose nodes are split into two subsets $V_1$ and $V_2$ consisting of $N_1$ and $N_2$ nodes, respectively. Furthermore, let $\mathbf{A}_{12}$ be the $N_1 \times N_2$ weighted adjacency matrix that encodes the mutual connections between the two parts as described in Theorem 1. Without loss of generality we can order the nodes of the network in such a way that the matrix has the structure

$$\mathbf{A}_{12} = \begin{pmatrix} \vec{a}_1 & \cdots & \vec{a}_m & \vec{0} & \cdots & \vec{0} \\ \vec{0} & \cdots & \vec{0} & \vec{0} & \cdots & \vec{0} \end{pmatrix}, \tag{14}$$

where we assume that $n$ nodes of the first subset are connected to $m$ nodes of the other subset and thus $\vec{a}_1, \ldots, \vec{a}_m \in \mathbb{R}^n$. According to Theorem 1, a perfect network isolator is found if $\text{rank}(\mathbf{A}_{12}) = 1$, i.e. if all vectors $\vec{a}_1, \ldots, \vec{a}_m$ are linearly dependent.

To investigate the robustness of network isolators, we start from a unit rank matrix rank $(\mathbf{A}_{12}) = 1$ and perturb it iteratively. In each step we choose one of the vectors $\vec{a}_i, i = 1, \ldots, m$ at random and perturb it according to $\vec{a}'_i = \vec{a}_i + \vec{e} \parallel \vec{a}_i \parallel$. The elements of the perturbation vector $\vec{e}$ are chosen uniformly at random from the interval $[-\beta, \beta]$, where $\beta$ is a small parameter, here $\beta = 0.05$.

The deviation of the perturbed matrix $\mathbf{A}_{12}$ from a unit rank matrix is quantified using its coherence statistics[38], Eq. (10),

$$\xi(\mathbf{A}_{12}) = 1 - \min_{i,j} \frac{\langle \vec{a}_i, \vec{a}_j \rangle}{\parallel \vec{a}_i \parallel \parallel \vec{a}_j \parallel},$$

where $\langle \cdot, \cdot \rangle$ denotes the standard scalar product on $\mathbb{R}^n$ and $\parallel \cdot \parallel$ denotes the $\ell^2$-norm. For a matrix $\mathbf{A}_{12}$ of unit rank we have $\xi(\mathbf{A}_{12}) = 0$ as all vectors are linearly dependent. For vectors deviating from linear dependence, the measure increases until it reaches its maximum value if two vectors are orthogonal with $\xi(\mathbf{A}_{12}) = 1$.

To create Fig. 2e, we repeated this process 1000 times starting from the perfect isolator shown in panel c. Edge weights were randomly chosen from a normal distribution $\mathcal{N}(10, 1)$ with mean $\mu = 10$ and variance $\sigma^2 = 1$ except for the isolator. The network isolator consists of four nodes in the left subgraph that are completely connected to four nodes in the other subgraph (see Fig. 1c). We select groups of four edges that are connected to a single node in one subgraph and to all four nodes in the other subgraph to have the same weight such that initially rank $(\mathbf{A}_{12}) = 1$. For each perturbed network, we evaluate $\xi(\mathbf{A}_{12})$ and the ratio of flow changes $R$ according to Eq. (3) averaged over all possible trigger links $\ell$ and distances $d$. For a perfect isolator, this ratio vanishes due to a vanishing numerator.

**Power grid data and cascade model.** Power grid data has been extracted from the open European energy system model PyPSA-Eur, which is fully available online[44]. The model includes the topology as well as the susceptance $b_\ell$ and the line rating $F_{i \to j}^{\max}$ for each high voltage transmission line in Europe. We consider the Scandinavian synchronous grid spanning Norway, Sweden, Finland and parts of Denmark. This grid is coupled to other synchronous grids (central European grid, British grid and Baltic grid) only via high voltage DC transmission lines. Power flow on these lines are actively controlled and can thus be considered constant, thus leading to constant real power injections at the coupling nodes. The Scandinavian grid has 269 nodes and 370 edges, counting multiple-circuit lines only once.

Cascading failures are simulated for fixed power injections $P_i$ for each node corresponding to an economic dispatch for the entire PyPSA-Eur model that includes a security margin given by the constraint $|F_{i \to j}| \le 0.8 \cdot F_{i \to j}^{\max}$. The cascade is triggered by the failure of a single line $(r, s)$ which is effectively removed from the grid. The simulation then proceeds step-wise; In each step, we first calculate the nodal phase angles $\vartheta_i$ and real power flows $F_{i \to j}$ for all nodes and lines, respectively,

by solving the continuity equation $P_i = \sum_j F_{i \to j}$ with $F_{i \to j} = A_{ij}(\vartheta_i - \vartheta_j)$. Then we check for overloads: Any line $(i, j)$ with $|F_{i \to j}| > F_{i \to j}^{\max}$ undergoes an emergency shutdown and is removed from the grid. The simulations are stopped when no further overload occurs or when the grid is disconnected.

Note that this mechanism for cascading failures is different from the cascading failure mechanism typically analysed in node capacity load models (see e.g. refs. [45,46]). The redistribution of nodal loads or flows after failures in such models is typically based on the neighbourhood of nodes, on shortest path betweenness measures or on other 'intelligent' redistribution schemes whereas the redistribution of flows after failures in linear flow networks or power grids studied using AC load flow analysis are given by the physical laws governing electrical networks. Furthermore, in most cases nodes—not edges—are assumed to fail, which is not the typical case in real power grids.

**Processing leaf data.** The leaf venation network is based on a microscopic recording of a leaf of the species *Bursera hollickii* provided by the authors of ref. [47]. Edge weights $A_{ij}$ are assumed to scale with the radius $r_{ij}$ of the corresponding vein $(i, j)$ as $A_{ij} \propto r_{ij}^4$ according to the Hagen–Poisseuille law, see ref. [28] for a detailed discussion. We used the radius in pixel scanned at a resolution of 6400 dpi.

**Parametrising network isolators with four edges.** Consider a network isolator that connects two vertices from one module with two vertices in the other module and consists of four edges in total (see Fig. 3a, right). Denote the weights of the four edges by $a_1, a_2, a_3, a_4$ and assume that we fix the overall available weight to build the network isolator. Including the rank conditions, the edge weights have to satisfy two constraints,

$$\sum_{\ell=1}^{4} a_\ell = \mathfrak{A} = 4$$

$$\text{rank}\left(\begin{pmatrix} a_1 & a_2 \\ a_3 & a_4 \end{pmatrix}\right) = 1 \Rightarrow a_1 \cdot a_4 = a_2 \cdot a_3,$$

thus leaving two degrees of freedom. We can then solve this set of equations for two variables and treat the remaining ones, $a_3, a_4$, as parameters that are varied independently:

$$a_1 = a_3 \frac{(\mathfrak{A} - a_4 - a_3)}{a_3 + a_4}, \quad a_2 = a_4 \frac{(\mathfrak{A} - a_4 - a_3)}{a_3 + a_4}.$$

For the simulations shown in Fig. 5c, d. we have set $\mathfrak{A} = 4$.

**Varying the degree of non-linearity.** To vary the degree of non-linearity systematically in Fig. 5, we first randomly assign 25% of the nodes to be identical sources and the remaining ones to be identical sinks and choose their value such that Eq. (3) is fulfilled. We then calculate the non-linear flows by combining Eq. (2) with the non-linear flows (Eq. (11)). For sources, we set $P_i = 0.09$ (bright line) initially and then systematically increase (decrease) the injections at all sources (sinks) by the factors 3.5, 6.0, 8.5, 11 (lines from light to dark) up until a maximum value of $P_i = 0.99$ is reached (black line) which corresponds to a maximum flow in the network of $|\tilde{F}|_{\max} = 0.89$.

## Data availability

The topology of the Scandinavian power grid, the Central European power grid and the British power grid have been extracted from the open European energy system model PyPSA-Eur[44], which is fully available online at https://doi.org/10.5281/zenodo.3886532. Leaf data was provided by the authors of ref. [47] and is available from the respective authors upon reasonable request.

## Code availability

Computer code will be made available at https://github.com/FNKaiser/Inhibiting_Failure_Spreading upon publication.

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

## Acknowledgements
We thank Tom Brown and Jonas Hörsch for help with the processing of power grid data, Eleni Katifori and Henrik Ronellenfitsch for providing the leaf data and Raissa D'Souza and Jürgen Kurths for valuable discussions. We gratefully acknowledge support from the Federal Ministry of Education and Research (BMBF grant no. 03EK3055B D.W.), the Helmholtz Association (via the joint initiative "Energy System 2050—a Contribution of the Research Field Energy" and grant no. VH-NG-1025 to D.W.). V.L.'s work was funded by the Leverhulme Trust Research Fellowship CREATE.

## Author contributions
D.W. conceived research and acquired funding. F.K., V.L. and D.W. designed research. F.K. carried out all numerical simulations, evaluated the results and designed all figures. All authors contributed to discussing the results and writing the paper.

## Funding

## Competing interests
The authors declare no competing interests.

## Additional information

                                                                                 9