## [Peer Review File · Nature Communications]

Reviewer #2 (Remarks to the Author):

The manuscript describes a study of cascading failures in interconnected complex networks such as power grids and proposes a method for controlling and limiting spread of such failures through alteration of connectivity at the interface of coupled networks. Authors show that for a class of linear dynamical systems defined by a conservation law the response of a network to small perturbation is determined by the Laplacian of that system. This allows authors to define particular inter-modular coupling for which failure of a link in one module (network) does not affect flows in other modules (networks). Authors test their approach on several synthetic and real-life networks, as well as discuss applicability of their approach in a nonlinear case.

I agree that the topology of networks plays an important role in controlling dynamical processes occurring on those networks, and thus I find the work presented in manuscript interesting and relevant for readers of Nature Communications. However, I have few questions that at this moment prevent me from recommending the manuscript for publication in current form.

1) My first concern regards applicability of presented approach in a nonlinear case. I appreciate the discussion and simulations presented in section "Network isolators beyond linear flow networks", however I wonder how those results connect with an earlier paper of some of the authors, "Dynamically induced cascading failures in power grids" (Nature Communications 2018), in which they stress the importance of looking at transient behavior after a failure, rather than at just evaluating steady-state solutions. This also connects to the discussion included in Supplement "Network isolators in non-linear systems", where authors consider a perturbation to nonlinear dynamics around a fixed point. It seems to me that authors discuss relatively simple cases of nonlinear dynamics, what might be an oversimplification of the studied problem and overextension of proposed methodology.

2) The issue that connects to my earlier question is nonlocality of failures observed in power outages. In this manuscript, as in their previous work, the authors demonstrate that probability of a link failure decreases monotonically with its distance from the original failed edge. This contradicts the observations of real power grid cascades, as well as numerical work presented in "Small vulnerable sets determine large network cascades in power grids" (Science 2017). I would like for the authors to comment on this point.

3) I am lacking a physical interpretation for the action performed by network isolators. In an earlier work by one of the authors, "Critical links and nonlocal rerouting in complex supply networks" (PRL 2016) it was shown that an impact of an edge in producing large-scale failure is connected to its redundant capacity, which meant presence of additional paths for rerouting the flow. It seems to me that proposed network isolators play just that role. If so, wouldn't they possibly introduce additional loops into the system, which might be difficult to control?

Minor issues:

1) Statement in the abstract "both weak and strong connections": from my perspective in network science literature "weak/strong" connections usually refer to links with low/high weights (eg. work of Granovetter). To me, what authors have in mind is sparse/dense connectivity. Authors discuss in general a weighted network, but the connectivity between layers is realized by a fraction c of nodes in each layer. This parameter (variable) is not discussed more broadly in the paper. I think a little bit more distinction is needed here, since a lot of papers discussing interlayer networks discuss the effects of increasing the number of nodes that are coupled between layers, while authors focus on the density of interlayer edges between a small set of nodes.

2) I would not agree that addition of edges is particularly "counterintuitive" (line 22) as a method for controlling cascading failures. Observation of this nature has been made e.g. in "Suppressing cascades of load in interdependent networks" (PNAS 2012). I would prefer for the authors to remove emotionally charged words and stick to science.

3) Figure 4c: I would like more explanation in the captions for this panel. I know there are people who start reading papers by looking at the figures and here more detailed description, and an

important fact that creation of a network isolator in this case requires adding nodes to the system, is needed. In general, this part of the figure was difficult to understand for me.

4) Sentence from lines 282-285. This sentence is really confusing to me each time I read it. I understand the idea, but I am losing my train of thoughts here.

5) Figure 5d. There are no straight lines on the plot, while captions mention "light blue, straight line". In general, I wonder how well this color scheme (light blue vs dark blue) is going to work e.g. in black-and-white print.

6) Supplementary Figure 4. I do not see failing link on panel C. Also, I do not understand why nodes are colored differently on panels B-D.

Reviewer #4 (Remarks to the Author):

The manuscript describes a study of a method of "network isolator" in preventing the spread of failures in flow networks. A theorem is presented on a sufficient condition for the connectivity between two or more network modules/communities to completely prevent a single link removal in one module from affecting flows in the other modules. Linear flows over random networks are used to illustrate how link/nodes modifications based on edge/vertex cut sets in a given network can prevent the spread of single link failure. Applications to power grids using the DC power flow model is presented, illustrating that such network modifications can prevent cascading failures from occurring. It is also shown that some results based on the DC power flow model approximately carry over to the AC model as well.

The idea of creating a "network isolator" to prevent cascading failures is interesting, and application to power-grid networks is relevant. However, I have a few major concerns that prevent me from recommending the manuscript for publication in Nature Communications in its current form:

1) Since many real applications involve nonlinear network systems, having some theoretical basis for the method's effectiveness on nonlinear systems is important. The authors do present two results on this front, but the condition under which the second extension (to "weakly" nonlinear systems) is valid is not very clear. Based on the materials presented in Supplementary Note 3, a key assumption seems to be that the change of the fixed point Δx_j is small, but this is different from what is meant by "weakly" coupled oscillators that can be modeled by Kuramoto model. The description in the main text is not clear about this, either. What is the condition on the nonlinear system under which a rank-1 "network isolator" is effective, and in what sense is it effective? For both types of extensions discussed in Supplemental Information, both the result and the condition for its validity should be clearly stated in the main text. The current description in the section "Network isolators beyond linear flow networks" is not adequate ("the existence of isolators" is ambiguous, and the statement for the second extension does not clearly say what the result is and what the condition is).

2) I understand that there are multiple choices of rank-1 matrices for A_{12} , even after the nodes to be involved are selected. This non-uniqueness naturally leads to a question about which of them is more/most effective in preventing a failure in one module to affect flow in another module in the case of nonlinear systems. Is there an theoretical answer to this question for the class of nonlinear systems for which a theory is presented? For applications, can the authors numerically investigate this question beyond just illustrating for a single choice?

3) Significant potential value of the method lies in the fact that it could prevent cascading failures involving multiple steps of failure propagation, but most of the manuscript focuses on a single step, which is what the linear theory addresses. The only result on cascading failures is the numerical result on the Scandinavian power grid for single triggering failure (Fig. 4). Could the authors present more solid evidence that creating "network isolators" can reduce the vulnerability of the power grid by showing statistics of cascading failures before and after introducing "network isolators" into the system?

Overall, I find the descriptions in many parts of the manuscript to be incomplete, inconsistent, and/or unclear. I have a number of specific comments, mostly on the presentation:

4) Fig. 1: In the caption, I would suggest providing a little more detail about the networks being shown and add a reference to Methods (or somewhere else) for more details.

5) Fig. 2a,b: What value of c was used? Define "6-regular graph" (is it used in the usual graph-theory sense?) and specify which 6-regular graph is used. In panel e, is the median and 25%/75% percentiles of $R(\ell)$ over ℓ also? For "a perfect network isolator", either fully specify what was used or refer to the Method section where it is specified.

6) Line 150: Define "distance".

7) Line 154: Clarify what is meant by "the remote part of the network $G' = O$ ". Is a "part" referring to a single multiple modules/communities?

8) The word "capacity" is used in Fig. 3 caption and other parts of the manuscript (including the SI), but for some readers (including myself) it sounds like the maximum link flows ($F^{\max}_{i \rightarrow j}$) used in the model of (multi-step) cascading failures in power grids. I would use something like "linear coefficient" or just "coefficient" for A_{ij} , which would avoid the potential confusion and also has the benefit of sounding more general.

9) Fig. 3: The authors seem to suggest that panels a - c defines the "recipes" for creating a "network isolator", but I cannot identify a clear instruction on how to systematically do that for a given network.

10) Line 172: The definition of "network isolator" is not clear. Does it refer to any A_{12} of rank one? I suggest making it clear what they mean by "network isolator" when the term is introduced for the first time in the main text.

11) Line 205: The "degree matrix" may not be clear. Is it "the diagonal matrix having node degrees as the diagonal elements"?

12) Fig. 4: The authors seem to suggest that the part shown in panel c is the part indicated by gray box in panels a and d, but the number of nodes in the boxes do not seem to match. Please clarify.

13) Eq. (10): It would be helpful have a sentence just after this equation to explain how this quantity is based on measuring the angle between the column vectors and that why $\xi = 0$ if and only if all the column vectors are parallel.

14) Line 247: Here and also in other places, a "grid" is used to refer to the system even when the discussion is more general than power grids. Please check.

15) Fig. 5d: The description in the caption is not clear as to the meaning of each curve shown.

16) Line 351: "inhibit the spreading completely" is only valid for linear flow networks. Every time a similar statement is made, care should be taken to phrase it accurately.

17) Line 354: The authors state that the method enable mitigating cascading failure while allowing for *an arbitrary degree* of connectivity between different parts of the system, but Fig. 2d suggests that network isolators are not so effective in the intermediate range of connectivity (as measured by μ), so that statement does not seem accurate in that case. Please clarify.

18) Line 406: If I understand correctly, $\xi(A_{12})$ reaches its maximum value of one when there exists two column vectors in A_{12} that are orthogonal (not just linearly independent). Please check.

19) Line 412: The description of the isolator is not clear. Did the authors mean that they chose "2 sets of four nodes (one set from each node group V_i , $i = 1,2$, and connected them to form of a complete bipartite graph"? If so, how were the nodes chosen? Randomly?

20) Methods: It took me a while to realize that details on the random network models used and the procedure used to produce the results in Fig. 2e are given in Methods, since no reference to the relevant Method sections are made in the main text. There is also a lot of overlap between the description of the procedure between the main text and the corresponding Method section. I suggest making sure it is clear in the main text and figure captions where to find further details.

21) Supplementary Fig. 6: The caption states, "We thus conclude that introducing the network isolator will not make the network more vulnerable compared to the network without the isolator." This may be misleading, since it is based solely on a single instance. Statistics are needed to support such a general claim.

22) Supplementary Fig. 7: The reference to Fig. 4 for the "recipes" does not seem to be correct. Is it supposed to be Fig. 3?

Response to Reviewers for:
Inhibiting failure spreading in complex networks

Franz Kaiser,^{1,2} Vito Latora,^{3,4,5} and Dirk Witthaut^{1,2}

¹*Forschungszentrum Jülich, Institute for Energy and
Climate Research (IEK-STE), 52428 Jülich, Germany*

²*Institute for Theoretical Physics, University of Cologne, Köln, 50937, Germany*

³*School of Mathematical Sciences, Queen Mary University of London, London E1 4NS, UK*

⁴*Dipartimento di Fisica ed Astronomia,*

Università di Catania and INFN, 95123 Catania, Italy

⁵*The Alan Turing Institute, The British Library, London NW1 2DB, UK*

REPLY TO REVIEWER 2

"The manuscript describes a study of cascading failures in interconnected complex networks such as power grids and proposes a method for controlling and limiting spread of such failures through alteration of connectivity at the interface of coupled networks. Authors show that for a class of linear dynamical systems defined by a conservation law the response of a network to small perturbation is determined by the Laplacian of that system. This allows authors to define particular inter-modular coupling for which failure of a link in one module (network) does not affect flows in other modules (networks). Authors test their approach on several synthetic and real-life networks, as well as discuss applicability of their approach in a nonlinear case.

I agree that the topology of networks plays an important role in controlling dynamical processes occurring on those networks, and thus I find the work presented in manuscript interesting and relevant for readers of Nature Communications. However, I have few questions that at this moment prevent me from recommending the manuscript for publication in current form."

→ We thank the reviewer for the in depth review of our paper. We are glad for the positive assessment of our work, and are grateful for the helpful comments, which we address below point by point.

Major comments

"1) My first concern regards applicability of presented approach in a nonlinear case. I appreciate the discussion and simulations presented in section "Network isolators beyond linear flow networks", however I wonder how those results connect with an earlier paper of some of the authors, "Dynamically induced cascading failures in power grids" (Nature Communications 2018), in which they stress the importance of looking at transient behavior after a failure, rather than at just evaluating steady-state solutions. This also connects to the discussion included in Supplement "Network isolators in non-linear systems", where authors consider a perturbation to nonlinear dynamics around a fixed point. It seems to me that authors discuss relatively simple cases of nonlinear dynamics, what might be an

oversimplification of the studied problem and overextension of proposed methodology.”

→ We thank the reviewer for this helpful comment. We agree that our framework has a greater applicability in linear systems where exact isolation is possible and believe that exact Theorems are much harder to obtain for the corresponding non-linear system or even the transient dynamics. Nevertheless, we also agree that it is an interesting question to what extent results carry on to nonlinear system beyond the example shown in Figure 5 of the manuscript. We therefore substantially extended this part in the revised version of the manuscript by numerically analysing non-linear systems and transient effects. In particular, we designed three new Figures to examine different aspects of network isolators in non-linear systems.

Inspired by the reviewer’s comment on Ref. [41], we examine transient amplitudes in the presence of network isolators in two different networks in Figure 2 of this revision file. To this end, we consider the second order Kuramoto model and simulate the failure of a single link (panels a,d). We first consider a network containing an imperfect isolator, i.e. an isolator where a single link is missing and the network where the isolator link (dotted) was added. We then monitor the largest (absolute) transient amplitude T after the failure and compare the amplitudes in the network without the isolator and the network containing an isolator. In most cases, we observe no significant change in the amplitudes. Considering only the largest changes in amplitudes, however, we observe a reduction in transient amplitude sizes in the networks containing an isolator – thus in the two networks considered here, we observe a smaller risk of transient overloads when network isolators are introduced into the networks. We also added this Figure to the Supplemental Material of the manuscript where it is now Supplemental Figure 11 and refer to the Figure in the section on non-linear systems in the main text.

In addition to that, we now systematically analyse how the degree of non-linearity affects the isolator’s ability to suppress flow changes. To this end, we calculate the flow changes after link failures with increasing non-linearity for different networks and distributions of sources and sinks in Figure 3. We then evaluate how the ratio of flow changes \tilde{R} , evaluated for the non-linearly calculated flow changes $\Delta\tilde{F}$, scales with the degree of non-linearity. We observe that even for moderate degrees of non-linearity, the network isolator still significantly suppresses perturbation spreading in the system. We added this figure to the Supplemental

Material in the revised version of the manuscript where it is now Supplementary Figure 10.

Finally, we added a third figure that discusses which choices of edge weights provide the strongest isolation effect after link failures in non-linear systems described by the Kuramoto model. To this end, we consider a network isolator consisting of four edges, vary the weights of the edges systematically and monitor the resulting non-linear flow changes. We added this Figure to the main text of the manuscript where it is now Figure 5 and extended the corresponding section and methods to discuss the robust design of isolators in non-linear systems in more detail (see Figure 4 of this document).

” 2) The issue that connects to my earlier question is nonlocality of failures observed in power outages. In this manuscript, as in their previous work, the authors demonstrate that probability of a link failure decreases monotonically with its distance from the original failed edge. This contradicts the observations of real power grid cascades, as well as numerical work presented in “Small vulnerable sets determine large network cascades in power grids” (Science 2017). I would like for the authors to comment on this point.”

→ We thank the reviewer for bringing up this interesting question. We would like to take the opportunity to briefly discuss the mechanisms involved in nonlocality of cascading failures as compared to the decay of flow changes with distance. The present manuscript and our previous work focus mainly on flow rerouting ΔF as a result of link failures (see e.g. colour code in Figures 1,3,5). In previous work (Strake et al., NJP, (5), 053009, 2019) we have shown that in irregular networks, flow rerouting is in fact determined by the rerouting distance between two links, not their topological distance. This can lead to highly non-local flow changes when considering the topological network distance or even the geographical distance between two links.

Furthermore, in the DC approximation, an overload occurs at line (i, j) if

$$|F_{i \rightarrow j} + \Delta F_{i \rightarrow j}| > F_{i \rightarrow j}^{\max}$$

Hence, not only the flow rerouting $\Delta F_{i \rightarrow j}$ matters, but also the prior loading of a line. If the ratio between prior loading and maximal loading, $F_{i \rightarrow j} / F_{i \rightarrow j}^{\max}$, is very close to unity, already a tiny flow change $\Delta F_{i \rightarrow j}$ can lead to an overload – thus potentially resulting in a highly

nonlocal cascade propagation. To illustrate this interplay between relatively localised flow changes and heterogeneously distributed flows we map out the first three steps of the cascade shown in Figure 4 in the main text in Figure 1 of this document. Although flow changes after the failures are relatively localised (b,d,f), the overall cascade propagation is non-local in terms of geographical and network distance (a,c,e). In particular, this non-locality is visible in the final cascade steps (see Figure 4b in the main text). Thus, relatively local flow redistribution leads to a non-local cascade propagation here. We added this Figure to the Supplemental Material as Supplementary Figure 12 and now explicitly mention the non-locality in the main text:

”To analyse how the relatively localised flow changes involved here lead to a non-local cascade, individual steps of the cascade are shown in Supplementary Figure 12.”

In addition to that, we illustrate the decay of non-linearly calculated flow changes with distance for different degrees of non-linearity Figure 4b of this document. Here, we clearly observe a reduction of flow changes with distance – even in the non-linear case.

In total, we hope that we convinced the reviewer that our results do not contradict the non-locality observed in power outages and analysed in the manuscript mentioned by the reviewer. In contrast, we hope that our findings on the locality of flow changes can widen our understanding of the non-locality involved in cascade propagation.

Finally, we would like to draw the reviewer’s attention to a second important aspect involved in the nonlocality of cascading outages in power grids. Reanalyses have demonstrated that nonlocality in power outages is often also related to voltage stability issues (see e.g. Venkatasubramanian et al., Bulk Power System Dynamics and Control, VI. Cortina d’Ampezzo, Italy, 685–721, 2004). Unfortunately, covering these effects requires more involved models that incorporate voltage dynamics (see e.g. Simpson-Porco et al., Nat. Comm. 7, 10790, 2016 or Schmietendorf et al., European Physical Journal Special Topics, 223, 2577, 2014) and go beyond the linear flow models introduced here. Nevertheless, we agree that it would be interesting to study to what extent the results presented here can be applied to more complex models which is, however, beyond the scope of the present work.

”3) I am lacking a physical interpretation for the action performed by network isolators. In an earlier work by one of the authors, “Critical links and nonlocal rerouting in complex supply networks” (PRL 2016) it was shown that an impact of an edge in producing large-scale

failure is connected to its redundant capacity, which meant presence of additional paths for rerouting the flow. It seems to me that proposed network isolators play just that role. If so, wouldn't they possibly introduce additional loops into the system, which might be difficult to control?"

→ We thank the reviewer for bringing up this previous work of ours. Indeed, network isolators may introduce additional loops into the system if they are a result of link addition. However, as we illustrate in Figure 3c, it is also possible to create network isolators by removing instead of adding new loops to the system. Nevertheless, the question of the "physical interpretation" of network isolators in relationship to the redundant capacity points to an interesting aspect.

In fact, the redundant capacity may be interpreted as an upper bound on the maximal flow that can be routed through alternative paths in case of a link failure and thus as a necessary condition on stability after the failure. A prediction of stability based on the redundant capacity works well if flow rerouting is dominated by one particular rerouting path or if the dominating paths are independent from each other, but can yield worse results if different important paths "interact". This is due to the fact that the redundant capacity does not incorporate the nodal potentials ϑ (see Eq.(1)) and thus the directionality and interaction of different paths (using an analogy from physics, they may be pictured as an incoherent approximation of coherent scattering phenomena, where the information of the interaction is lost). The loops introduced by network isolators may be interpreted to play exactly that role: Due to the high degree of symmetry, different rerouting paths interact coherently and thus balance one another, resulting in an effective barrier for flow rerouting. We hope that this explanation clarifies the physical interpretability of network isolators and its relationship towards redundant capacities.

To point to possible physical interpretations of network isolators, we added the following statement to the final discussion in the main text:

"Furthermore, our results show that algebraic properties of networks can have striking effects on their function and robustness – depending on the type of flow model. Similar effects are not present in simple models where flows are rerouted along the shortest paths only [4,9], but they can become essential in physical supply network models where various paths contribute and interact in a non-trivial way."

Minor comments:

"1) Statement in the abstract "both weak and strong connections": from my perspective in network science literature "weak/strong" connections usually refer to links with low/high weights (eg. work of Granovetter). To me, what authors have in mind is sparse/dense connectivity. Authors discuss in general a weighted network, but the connectivity between layers is realized by a fraction c of nodes in each layer. This parameter (variable) is not discussed more broadly in the paper. I think a little bit more distinction is needed here, since a lot of papers discussing interlayer networks discuss the effects of increasing the number of nodes that are coupled between layers, while authors focus on the density of interlayer edges between a small set of nodes."

→ We thank the reviewer for making us aware of this misleading statement in the abstract. In general, our intention was not to discuss interlayer networks in this manuscript and we are sorry for our confusing statement. We rephrased the corresponding sentence such that it now reads

" In this article, we introduce a general framework to analyse the spreading of failures in complex networks and demonstrate that not only decreasing but also increasing the connectivity of the network can be used to contain damages."

"2) I would not agree that addition of edges is particularly "counterintuitive" (line 22) as a method for controlling cascading failures. Observation of this nature has been made e.g. in "Suppressing cascades of load in interdependent networks" (PNAS 2012). I would prefer for the authors to remove emotionally charged words and stick to science."

→ We thank the reviewer for making us aware of similar findings in the context of sandpile models. We removed the word "counterintuitive" from the respective sentence and decided to cite the manuscript suggested by the reviewer.

" 3) Figure 4c: I would like more explanation in the captions for this panel. I know there are people who start reading papers by looking at the figures and here more detailed description, and an important fact that creation of a network isolator in this case requires

adding nodes to the system, is needed. In general, this part of the figure was difficult to understand for me.”

→ We are sorry that this was unclear and thus extended the Figure legend as follows:

”A small modification of the grid enables the construction of a network isolator: adopting the recipe presented in Figure 3a, we select two nodes (left) that are further split up into two separate nodes each which are mutually connected via a network isolator by adding four edges (right, green).”

We hope that panel c is now easier to understand.

” 4) Sentence from lines 282-285. This sentence is really confusing to me each time I read it. I understand the idea, but I am losing my train of thoughts here.”

→ We excuse for the misleading statement and rephrased it. We hope that it is now easier to understand. The statement now reads:

”In the original grid layout, the modules are weakly connected, thus failure spreading between these modules is reduced – but it is possible.”

” 5) Figure 5d. There are no straight lines on the plot, while captions mention “light blue, straight line”. In general, I wonder how well this color scheme (light blue vs dark blue) is going to work e.g. in black-and-white print.”

→ We thank the reviewer for pointing out this typo to us and corrected it. The legend now reads:

d We compare the median absolute flow changes, calculated using the non-linear load flow (Eq. (11)), after the failure of the link in the initial grid (dashed lines, a) and the modified grid (dotted lines, c). Whereas the flow changes in the lower module of the power grid (dark blue nodes) stay approximately the same after the grid modification (dark blue lines), they are significantly reduced in the grid’s upper module (light blue nodes) that is shielded by the network isolator (light blue lines).”

The colors shown here (light blue, dark blue) were chosen from a sequential colormap

where the lightness values increase monotonically. For this reason, we believe that the color scheme will also work in black-and-white print.

” 6) Supplementary Figure 4. I do not see failing link on panel C. Also, I do not understand why nodes are colored differently on panels B-D.”

→ We modified the node color such that nodes are colored black in all panels and chose a different failing link in panel c that is more clearly visible.

Figure 1. **Non-locality of cascade propagation and decay of flow changes** We illustrate the first three steps of the cascade in the Scandinavian power grid shown in Figure 4 in the main text for the grid without a network isolator. **a** Line loading in the Scandinavian grid prior to the initial failure with the initially failing link highlighted. Note that line loading is heterogeneously distributed within the network. **b** Relative flow changes $|\Delta F / F_{\text{fail}}^{(0)}|$ with respect to the flows on the initially failing link $F_{\text{fail}}^{(0)}$. The flow changes clearly decay with distance from the failing link. **c** Line loading after the initial failure: also flow changes decay with distance, the next failing link is relatively far apart from the initially failing link when considering the geographic or geodesic network distance. **d** Relative flow changes after the failure of the link shown in c. Again, the flow changes are localised. **e** Line loading after the failure of both links shown in panels a and c. The next failure is closer to the failing link shown in c, but even farther apart from the initially failing link, leading to an overall non-local cascade of failures. **f** Again, relative flow changes are strongly localised.

Figure 2. **Transient amplitudes are slightly reduced in the presence of network isolators.** **a** We analyse a network consisting of two modules that are connected via three links and add a fourth link (dotted) to create a network isolator. We randomly assign 25% of the nodes to be generator nodes (squares) and the remaining ones to be load nodes (triangle). We then simulate the removal of a single link (red) and monitor the corresponding response in the dynamic nonlinear system described by the second order Kuramoto model. **b** Non-linear dynamics of the flows in the upper module after the failure of a single link at time zero (dotted, vertical line) in the network before (straight lines) and after the addition of the isolator link (dotted lines). We monitor the maximum Amplitude T of the transient dynamics comparing the fixed point before and after the failure (inset). **c** To analyse the impact of network isolators on transient overloads, we compare the transient amplitudes before ($T_{\text{no isolator}}$) and after (T_{isolator}) constructing the isolator in the upper module for all possible link failures in the lower module. In most cases, the transient amplitudes stay the same after introducing the network isolator as confired by the mean close to zero (black, dashed line). However, evaluating only the 95% changes in amplitudes with the largest changes in magnitudes (dotted line), we observe a significant shift towards positive values indicating a reduced risk of transient overloads when network isolators are present. **d-f** The result is confirmed by performing the same analysis for a different network containing a larger network isolator.

Figure 3. **Network isolation effect persists for non-linear flows.** **a** We consider the six-regular graph shown in Figure 1f and simulate 50 different initial conditions where we randomly assign 25% of the nodes to be sources with $P_i = 0.9 \cdot \delta$ and the sinks correspondingly to balance the sources. Here, δ is a prefactor tuning the degree of non-linearity in the non-linear flows $\tilde{F}_{i \rightarrow j} = A_{i \rightarrow j} \sin(\vartheta_i - \vartheta_j)$. **b** For each initial condition, we analyse the maximum flow in the network $|F_{\max}|$ as an indicator of non-linearity for different degrees of non-linearity δ . **c** We then evaluate the ratio \tilde{R} of non-linear flow changes which is obtained from Eq.(8) by replacing the flow changes ΔF by their non-linear counterpart and averaging over all distances and trigger links in the left module. To examine to what extent network isolators prevent perturbation spreading from the left module to the right module, we plot this ratio against the non-linearity factor. With increasing degree of non-linearity, there is no longer exact isolation, i.e. $R = 0$, but a strong shielding effect persists. **d-i** We perform the same type of analysis for two three-regular graphs (d) and two random graphs $G(16, 0.3)$ (g) connected via network isolators and observe a similar scaling of the ratio \tilde{R} with the non-linearity factor. Shaded regions indicate half a standard deviation evaluated over all repetitions for all plots.

Figure 4. **Robust design of network isolators in weakly non-linear systems.** **a** We analyse a network consisting of two modules that are connected via three links and add a fourth link (dotted) to create a network isolator. We randomly assign 25% of the nodes to be generator nodes (squares) and the remaining ones to be load nodes (triangle). We then simulate the removal of a single link (red) and monitor the corresponding response in the dynamic (b,c) and static (d-f) nonlinear system described by the second order Kuramoto model. **b** We analyse the effect of edge distance (x-axis) and non-linearity on the isolation effect by comparing the flow changes $|\Delta F(\ell)|$ after the failure of a link in the right module (top) and the left, isolated module (bottom) with an increasing degree of non-linearity. As expected, flow changes in the right module are lowest for a weakly non-linear system (bright line) and increase with the non-linearity. However, a strong isolation effect persists even for the highest degree of non-linearity considered here (black lines). **c** To systematically study the isolator’s shielding capabilities, we fix the overall available edge weight of the four edges building the isolator to $\sum_i a_i = 4$ and systematically scan over the remaining degrees of freedom, measuring the performance by means of the mean logarithmic flow changes $\langle \log_{10}(|\Delta F|) \rangle$. **d** We evaluate the available worst-case $N - 1$ weight, i.e. the overall edge weight connecting the two modules after the failure of a single link in the isolator for the same set of edge weights as in c.

REPLY TO REVIEWER 4

"The manuscript describes a study of a method of "network isolator" in preventing the spread of failures in flow networks. A theorem is presented on a sufficient condition for the connectivity between two or more network modules/communities to completely prevent a single link removal in one module from affecting flows in the other modules. Linear flows over random networks are used to illustrate how link/nodes modifications based on edge/vertex cut sets in a given network can prevent the spread of single link failure. Applications to power grids using the DC power flow model is presented, illustrating that such network modifications can prevent cascading failures from occurring. It is also shown that some results based on the DC power flow model approximately carry over to the AC model as well. The idea of creating a "network isolator" to prevent cascading failures is interesting, and application to power-grid networks is relevant. However, I have a few major concerns that prevent me from recommending the manuscript for publication in Nature Communications in its current form:"

→ We thank the reviewer for this positive assessment of our work and are grateful for the helpful comments which we address below point by point.

Major comments

"1) Since many real applications involve nonlinear network systems, having some theoretical basis for the method's effectiveness on nonlinear systems is important. The authors do present two results on this front, but the condition under which the second extension (to "weakly" nonlinear systems) is valid is not very clear. Based on the materials presented in Supplementary Note 3, a key assumption seems to be that the change of the fixed point Δx_j is small, but this is different from what is meant by "weakly" coupled oscillators that can be modeled by Kuramoto model. The description in the main text is not clear about this, either. What is the condition on the nonlinear system under which a rank-1 "network isolator" is effective, and in what sense is it effective? For both types of extensions discussed in Supplemental Information, both the result and the condition for its validity should be clearly stated in the main text. The current description in the section "Network isolators

beyond linear flow networks” is not adequate (“the existence of isolators” is ambiguous, and the statement for the second extension does not clearly say what the result is and what the condition is).”

→ We thank the reviewer for this helpful comment and agree that this was misleading in the original manuscript. We therefore renamed the subsection in the Supplemental Material named “Approximate isolation for weakly non-linear systems” to “Approximate isolation for diffusively coupled non-linear oscillator networks”. We also revised the section such that the important assumptions of a diffusive coupling as assumed for example in the Kuramoto model and the small change in the fixed point Δx_j after the perturbation are now more clearly presented as crucial assumptions to apply a linear response theory. In the revised version, we also stress once more that the result is only approximately valid. This is also why we refrain from presenting it in a more formal form - e.g. a Corollary.

To study the interesting question of when a network isolator in non-linear systems is effective, we extended the manuscript in several regards. First, we revised and extended the introductory paragraph of the corresponding section in the main text where we refer to network isolators in non-linear systems such that it now reads:

”The concept of network isolators has been established for linear flow networks, but can be extended in two ways. (1) We can rigorously prove that network isolators determine the response to structural damages for a class of non-linear networked dynamical systems with diffusive coupling. More precisely, the isolator effect is still rigorously valid if the dynamics of a node i depends on the state of the other nodes x_j only through the term $f_i(\sum_j L_{ij}x_j)$, where \mathbf{L} is the Laplacian and the function f_i satisfies $f_i(0) = 0$, but is arbitrary otherwise (see Supplementary Note 3, Corollary 2). (2) For many non-linear systems of practical importance, the impact of failures or perturbations is well described by a linearisation around an equilibrium or limit cycle (see Ref. [31]) for which an approximate isolation can be achieved (see Supplementary Note 3, subsection 4). ”

Second, we have added an entirely new paragraph and a new Figure 5 (see Figure 4 of this document) where we systematically investigate the impact of non-linearity for a prototypical nonlinear interaction term $\tilde{F}_{i \rightarrow j} = A_{ij} \cdot \sin(\theta_i - \theta_j)$. This term corresponds to the celebrated Kuramoto model that naturally reduces to the linear flow model if phase differences between neighbouring nodes are small. We describe this Figure in detail in response to the next

comment by the reviewer and added the following explanation when introducing the model in the main text:

”To systematically analyse how non-linearity affects failure spreading through network isolators we first consider a natural extension of the linear flows in Eq. (1), replacing the linear coupling by its sinusoidal counterpart

$$\tilde{F}_{i \rightarrow j} = A_{ij} \cdot \sin(\vartheta_i - \vartheta_j), \quad (1)$$

which yields the well-known Kuramoto model [42,43]. If phase differences between neighbouring vertices are small, one can expand the sine function as $\sin(\vartheta_i - \vartheta_j) = (\vartheta_i - \vartheta_j) + \mathcal{O}((\vartheta_i - \vartheta_j)^3)$ (see Supplementary Note 1). Hence, our previous result remain valid to linear order, whereas a higher degree of non-linearity may gradually weaken the effects. In particular, the effectiveness of a network isolator depends on the relative load of the edges $|\tilde{F}_{i \rightarrow j}|/A_{ij}$. We study this numerically by increasing the injections P_i at all nodes proportionally, thus increasing the relative edge loads and the importance of the non-linearity of the sine function.”

Finally, we added several Figures to the Supplemental Material. We systematically study the effect of non-linearity on network isolators for different networks and settings in a new Supplementary Figure 10 (Figure 2 of this document). To this end, we randomise the power injections in a network with 25% of the nodes being producers and 75% being consumers and then randomly swap their positions for 50 repetitions. For each repetition, we systematically increase the power injections which results in a systematic increase of the non-linear flows in the network. We then simulate all possible link failures in one module and evaluate the resulting ratio of non-linear flow changes to study the effectiveness of network isolators in non-linear systems. We then repeat this procedure for three different network topologies. Furthermore, we analyse the effect of network isolators on transient amplitudes after link failures in networks coupled dynamically via the second order Kuramoto model in Supplementary Figure 11.

”2) I understand that there are multiple choices of rank-1 matrices for for A_{12} , even after the nodes to be involved are selected. This non-uniqueness naturally leads to a question about which of them is more/most effective in preventing a failure in one module to affect flow in

another module in the case of nonlinear systems. Is there an theoretical answer to this question for the class of nonlinear systems for which a theory is presented? For applications, can the authors numerically investigate this question beyond just illustrating for a single choice?"

→ We thank the reviewer for this helpful comment. As pointed out correctly by the reviewer, any choice of a rank-1 matrix will yield the same result for linear flow networks by virtue of Theorem 1: Flow changes are completely suppressed independently of the choice and every realisation of a network isolator has the same effectiveness.

Nevertheless, to systematically analyse which kind of network isolator provides the best possible shielding for a non-linear system, we did additional numerical studies for the sinusoidal flows described by the Kuramoto model. We focused on a network isolator connecting two pairs of nodes via four edges with weights a_1, a_2, a_3, a_4 and fixed the overall available weight to build the network isolator. Using this set-up, we are left with two degrees of freedom which we can systematically analyse. For a given degree of non-linearity, we then analysed which choice of edge weights results in the lowest non-linear flow changes in the module shielded by the isolator averaged over all possible trigger links in the other module. Furthermore, we compare this to the worst-case available $N - 1$ weight, i.e. the overall edge weight connecting the two modules if the edge in the network isolator with the largest weight fails. Comparing the two metrics, we can analyse the trade-off between the best possible shielding provided by the isolator, on the one hand, and the risk in losing transport capacity between the two modules after a potential link failure in the isolator on the other hand. In Figure 4 of this document we present the results for a given network. We find that network isolators with strongly heterogeneous edge weights a_1 and a_2 inhibit failure spreading the most in the weakly non-linear system under consideration. However, the most symmetric choice with $a_i = 1, i \in \{1, 2, 3, 4\}$ yields the highest the available $N - 1$ weight, while still inhibiting failure spreading relatively strongly. We added this Figure to the revised version of the manuscript where it is now Figure 5 and significantly extended the discussion of non-linear systems and Methods with respect to this important aspect.

" 3) Significant potential value of the method lies in the fact that it could prevent cascading failures involving multiple steps of failure propagation, but most of the manuscript focuses on a single step, which is what the linear theory addresses. The only result on cascading

failures is the numerical result on the Scandinavian power grid for single triggering failure (Fig. 4). Could the authors present more solid evidence that creating "network isolators" can reduce the vulnerability of the power grid by showing statistics of cascading failures before and after introducing "network isolators" into the system?"

→ We thank the reviewer for raising the important question to what extent network isolators can prevent cascade propagation in general. First, we would like to comment on the results on cascade propagation shown in Figure 4: There are actually multiple steps involved in the cascade propagation in this example, even if the cascade is initially triggered by a single link. We decided to only show the final result of the cascade in the original version of the manuscript to keep the presentation brief even if multiple steps are involved in the cascade propagation. To clarify this in the revised version of the manuscript, we added a new Supplementary Figure 12 (1 of this document) which shows the first cascade steps in detail.

To systematically analyse the impact network isolators have on cascade propagation, we study two synthetic networks. We stick to synthetic networks here for two reasons: 1) A systematic study requires to tune the networks whereas real-world grids provide only single instances. 2) Cascading failures occur rarely for realistic dispatches of the energy system, such that a sufficient, convincing statistics is not available.

For the two topologies we then randomize the positions of producers and consumers and simulate the failure of any possible trigger link in one of the modules and monitor the size of the resulting cascade of failures. We then repeat this procedure many times for different placement of producers and consumers to obtain a statistics of cascades. Finally, we compare the cascade sizes in the other module that does not contain the trigger link for one network that contains a network isolator and another one where the isolator subgraph is slightly modified. As a result, we observe that cascade sizes in the module shielded by the isolator are significantly reduced and thus conclude that network isolators have the ability of strongly reducing the vulnerability of the power grid (see Figure 3).

We added this Figure to the Supplemental material of the revised version of the manuscript where it is now Supplemental Figure 9. Furthermore, we discuss the new Figure in the main text at the end of the section on cascade propagation as follows:

"To demonstrate that network isolators effectively suppress cascade propagation for dif-

ferent networks and initial failure patterns, we evaluate the statistics of cascade sizes in networks with and without network isolators (see Supplementary Figure 9).”

Minor comments

” Overall, I find the descriptions in many parts of the manuscript to be incomplete, inconsistent, and/or unclear. I have a number of specific comments, mostly on the presentation.”

”4) Fig. 1: In the caption, I would suggest providing a little more detail about the networks being shown and add a reference to Methods (or somewhere else) for more details.”

→ We updated the Figure caption to now clearly indicate where to find further information on the graphs used here. The caption now reads:

”See Methods and caption of Figure 2 for further information on the graphs used here.”

”5) Fig. 2a,b: What value of c was used? Define ”6-regular graph” (is it used in the usual graph-theory sense?) and specify which 6-regular graph is used. In panel e, is the median and 25%/75% percentiles of $R(\ell)$ over ℓ also? For ”a perfect network isolator”, either fully specify what was used or refer to the Method section where it is specified.”

→ In fact ”six-regular graph” is to be understood in the graph theoretical sense. We use the six-regular graph shown in Figure 1c to exclude any effect of heterogeneity in connectivity on failure spreading and therefore designed a graph that is exclusively six-regular, but contains a network isolator. We clarified this in the revised version of the manuscript by adding the statement

”Note that all nodes in the graph including those in the network isolator have degree equal to six, which allows us to exclude any potential impact of heterogeneity in the degree on failure spreading in this case.”

to the caption and added the value of $c = 0.2$ used in the other panels. The six-regular graph used here and shown in Figure 1c has 48 nodes. Unfortunately, we are not aware of any name for this particular graph. Finally, we now reference the Methods when discussing panel e.

"6) Line 150: Define "distance"."

→ We added a paragraph describing the definition of the notion of distance used throughout the manuscript to the 'Methods' section and changed the corresponding sentence in line 150 such that it now reads:

"which gives the magnitude of flow changes averaged over all edges $(i, j) \in G'$ at a given distance d to the edge ℓ (see Methods for details on the notion of distance used here)."

"7) Line 154: Clarify what is meant by "the remote part of the network $G' = O$ ". Is a "part" referring to a single multiple modules/communities?"

→ Yes, "part" is referring to the modules or communities of the network in this case. We are sorry for the confusion caused and clarified this statement in the revised version of the manuscript such that it now reads

"between the module of the network $G' = O$ without initial failures and the module $G' = S$ containing the failing edge ℓ ."

"8) The word "capacity" is used in Fig. 3 caption and other parts of the manuscript (including the SI), but for some readers (including myself) it sounds like the maximum link flows ($F_{i \rightarrow j}^{max}$) used in the model of (multi-step) cascading failures in power grids. I would use something like "linear coefficient" or just "coefficient" for A_{ij} , which would avoid the potential confusion and also has the benefit of sounding more general. "

→ We thank the reviewer for pointing out this potential source of confusion and agree with it. In the revised version of the manuscript, we now refer to A_{ij} by the more common name "weight".

"9) Fig. 3: The authors seem to suggest that panels a - c defines the "recipes" for creating a "network isolator", but I cannot identify a clear instruction on how to systematically do that for a given network. "

→ We extended the discussion of the Figure in the corresponding section. It now reads:

”For a given graph these recipes may thus be applied as follows: 1) Identify modules of the graph that are weakly connected to one another as measured by a low vertex cut or edge cut of the vertices or edges connecting them. 2) Depending on the target – e.g. whether building new edges or vertices is costly or, on the other hand, a minimum connectivity between the modules is required after the modification – identify the optimal strategy to achieve a complete bipartite connectivity between the modules by adding or removing vertices and edges. Here, the recipes shown in the Figure may be applied directly if the prior connectivity has the indicated edge or vertex cuts. 3) Tune the edge weights such that $\text{rank}(\mathbf{A}_{12}) = 1$ is achieved, i.e. a network isolator is realised..”

”10) Line 172: The definition of ”network isolator” is not clear. Does it refer to any A_{12} of rank one? I suggest making it clear what they mean by ”network isolator” when the term is introduced for the first time in the main text. ”

→ Yes, it refers to any subgraph connecting the modules such that A_{12} has unit rank. We clarified this in line 172 as follows:

”We will explain the concept of *network isolators* and provide a rigorous definition in the next section.”

To further clarify what is meant by the term ”network isolator” we added the following sentence to the end of Theorem 1:

”The subgraph connecting the two modules is referred to as a *network isolator*.”

” 11) Line 205: The ”degree matrix” may not be clear. Is it ”the diagonal matrix having node degrees as the diagonal elements”? ”

→ In fact, this refers to the usual definition of the degree matrix. We clarified this in the revised manuscript by adding the statement:

” \mathbf{D}_1 and \mathbf{D}_2 are the degree matrices of these mutual connections, i.e. the matrices containing the nodes’ weighted degrees on the diagonals.”

” 12) Fig. 4: The authors seem to suggest that the part shown in panel c is the part

indicated by gray box in panels a and d, but the number of nodes in the boxes do not seem to match. Please clarify.”

→ We revised the Figure such that the nodes are now highlighted in panels a and d and adjusted the figure caption in the revised version of the manuscript such that it more clearly describes what is shown in the corresponding panel. The caption now reads

”agnification of the grid structure in Eastern Norway (grey box, a). A small modification of the grid enables the construction of a network isolator: adopting the recipe presented in Figure 3a, we select two nodes (left) that are further split up into two separate nodes each which are mutually connected via a network isolator by adding four edges (right, green).”

” 13) Eq. (10): It would be helpful have a sentence just after this equation to explain how this quantity is based on measuring the angle between the column vectors and that why $\xi = 0$ if and only if all the column vectors are parallel. ”

→ We agree that this provides helpful additional insight and thus added the following statement after Eq. (10):

” Note that the latter expression $\cos(\angle \vec{a}_i, \vec{a}_j) = \frac{\langle \vec{a}_i, \vec{a}_j \rangle}{\|\vec{a}_i\| \|\vec{a}_j\|}$ may also be interpreted via the angle between the matrix columns \vec{a}_i and thus $\xi(\mathbf{A}_{12})$ approaches a value of unity if all columns are parallel.”

” 14) Line 247: Here and also in other places, a ”grid” is used to refer to the system even when the discussion is more general than power grids. Please check. ”

→ We changed the language as suggested by the reviewer. In the revised version of the manuscript, we use the term ”grid” only when referring to power grids and otherwise stick to the more general term ”network”.

” 15) Fig. 5d: The description in the caption is not clear as to the meaning of each curve shown.”

→ We revised and extended the figure caption to clarify. The new description of panel d

reads:

” **d** We compare the median absolute flow changes, calculated using the non-linear load flow (Eq. (11)), after the failure of the link in the initial grid (dashed lines, a) and the modified grid (dotted lines, c). Whereas the flow changes in the lower module of the power grid (dark blue nodes) stay approximately the same after the grid modification (dark blue lines), they are significantly reduced in the grid’s upper module (light blue nodes) that is shielded by the network isolator (light blue lines).”

” 16) Line 351: *”inhibit the spreading completely” is only valid for linear flow networks. Every time a similar statement is made, care should be taken to phrase it accurately.* ”

→ We are sorry for the misleading statement and revised it such that it now reads:

”We have demonstrated that an even stronger effect can be created by certain subgraphs called isolators, which inhibit the spreading of failures in linear systems completely. ”

” 17) Line 354: *The authors state that the method enable mitigating cascading failure while allowing for *an arbitrary degree* of connectivity between different parts of the system, but Fig. 2d suggests that network isolators are not so effective in the intermediate range of connectivity (as measured by μ), so that statement does not seem accurate in that case. Please clarify.* ”

→ In fact, building a network isolator is possible at an arbitrary degree of connectivity. Figure 2d does not refer to network isolators – they suppress failure spreading completely such that $R = 0$ by virtue of Theorem 1. The intermediate range of connectivity here refers to the graph ensemble created by randomly adding links as discussed also the legend to the Figure and the Methods section. Only Figure 2e actually discusses network isolators. We are sorry for the confusion caused here and clarified this in the revised version of the manuscript by swapping the order of panels c and d in the Figure, such that the scaling of the flow ratio with interconnectivity is now discussed immediately after introducing the graph ensemble with strong or weak connectivity. Only then we now discuss aspects related to network isolators in the revised Figure caption. We hope this clarified the issue raised by the reviewer.

” 18) Line 406: If I understand correctly, $\xi(A_{12})$ reaches its maximum value of one when there exists two column vectors in A_{12} that are orthogonal (not just linearly independent). Please check. ”

→ We thank the reviewer for pointing out this wrong statement to us and corrected it as suggested by the reviewer.

” 19) Line 412: The description of the isolator is not clear. Did the authors mean that they chose ”2 sets of four nodes (one set from each node group $V_i, i = 1, 2$, and connected them to form of a complete bipartite graph”? If so, how were the nodes chosen? Randomly? ”

→ We revised the corresponding part of the ”Methods” section such that it now reads:

” Edge weights were randomly chosen from a normal distribution $\mathcal{N}(10, 1)$ with mean $\mu = 10$ and variance $\sigma^2 = 1$ except for the isolator. The network isolator consists of four nodes in the left subgraph that are completely connected to four nodes in the other subgraph (see Fig. 1c). We select groups of four edges that are connected to a single node in one subgraph and to all four nodes in the other subgraph to have the same weight such that initially $\text{rank}(\mathbf{A}_{12}) = 1$.”

” 20) Methods: It took me a while to realize that details on the random network models used and the procedure used to produce the results in Fig. 2e are given in Methods, since no reference to the relevant Method sections are made in the main text. There is also a lot of overlap between the description of the procedure between the main text and the corresponding Method section. I suggest making sure it is clear in the main text and figure captions where to find further details. ”

→ We thank the reviewer for making us aware of this lack of details and agree that this requires revision. In the revised manuscript, we updated the Figure caption with a reference to the Methods section and a reference to the coherence parameter:

”e The ratio of flow changes R , now averaged over all possible trigger links ℓ and distances d , vanishes for a network isolator described by the condition $\xi(\mathbf{A}_{12}) = 0$ and increases algebraically with the coherence parameter ξ (cf. Eq. (10)) when perturbed (see Methods

for details on the simulation).”

Furthermore, we shortened the description of the exact methodology applied to perturb the adjacency matrix in the main text and now refer to the corresponding Methods section for further explanation. The revised section now reads:

”Since most real world examples of networks do not contain perfect network isolators, we have studied the robustness of a network isolator against modifications of the topology. Starting from a unit rank matrix, we perturb the adjacency matrix \mathbf{A}_{12} iteratively (see Methods for details). The deviation of the perturbed matrix \mathbf{A}_{12} from a unit rank matrix is then quantified using its coherence statistics ”

” 21) Supplementary Fig. 6: The caption states, ”We thus conclude that introducing the network isolator will not make the network more vulnerable compared to the network without the isolator.” This may be misleading, since it is based solely on a single instance. Statistics are needed to support such a general claim. ”

→ We thank the reviewer for pointing us to this misleading statement. We decided to tone down the message as we agree that statistics are needed to support the general claim. The new caption now refers solely to this single instance of simulation and reads

”We observe that both, the failure within the isolator (panel b) as well as a failure in the initial grid in close proximity to the location where the isolator is constructed (panel a) yield a similar effect. In this case, the network’s vulnerability is thus not increased by including the network isolator. However, a failure in the isolator may potentially affect the whole network.”

Nevertheless, we would like to highlight our new Figure 3 (Supplementary Figure 9 in the manuscript) on cascade statistics to the reviewer. The figure may be interpreted as a demonstration that network isolators do in fact not make the network more vulnerable in terms of the size of cascades.

” 22) Supplementary Fig. 7: The reference to Fig. 4 for the ”recipes” does not seem to be correct. Is it supposed to be Fig. 3? ”

→ We thank the reviewer for pointing out this typo and corrected it.

Figure 1. **Non-locality of cascade propagation and decay of flow changes** We illustrate the first three steps of the cascade in the Scandinavian power grid shown in Figure 4 in the main text for the grid without a network isolator. **a** Line loading in the Scandinavian grid prior to the initial failure with the initially failing link highlighted. Note that line loading is heterogeneously distributed within the network. **b** Relative flow changes $|\Delta F / F_{\text{fail}}^{(0)}|$ with respect to the flows on the initially failing link $F_{\text{fail}}^{(0)}$. The flow changes clearly decay with distance from the failing link. **c** Line loading after the initial failure: also flow changes decay with distance, the next failing link is relatively far apart from the initially failing link when considering the geographic or geodesic network distance. **d** Relative flow changes after the failure of the link shown in c. Again, the flow changes are localised. **e** Line loading after the failure of both links shown in panels a and c. The next failure is closer to the failing link shown in c, but even farther apart from the initially failing link, leading to an overall non-local cascade of failures. **f** Again, relative flow changes are strongly localised.

Figure 2. **Network isolation effect persists for non-linear flows.** **a** We consider the six-regular graph shown in Figure 1f and simulate 50 different initial conditions where we randomly assign 25% of the nodes to be sources with $P_i = 0.9 \cdot \delta$ and the sinks correspondingly to balance the sources. Here, δ is a prefactor tuning the degree of non-linearity in the non-linear flows $\tilde{F}_{i \rightarrow j} = A_{i \rightarrow j} \sin(\vartheta_i - \vartheta_j)$. **b** For each initial condition, we analyse the maximum flow in the network $|F_{\max}|$ as an indicator of non-linearity for different degrees of non-linearity δ . **c** We then evaluate the ratio \tilde{R} of non-linear flow changes which is obtained from Eq.(8) by replacing the flow changes ΔF by their non-linear counterpart and averaging over all distances and trigger links in the left module. To examine to what extent network isolators prevent perturbation spreading from the left module to the right module, we plot this ratio against the non-linearity factor. With increasing degree of non-linearity, there is no longer exact isolation, i.e. $R = 0$, but a strong shielding effect persists. **d-i** We perform the same type of analysis for two three-regular graphs (d) and two random graphs $G(16, 0.3)$ (g) connected via network isolators and observe a similar scaling of the ratio \tilde{R} with the non-linearity factor. Shaded regions indicate half a standard deviation evaluated over all repetitions for all plots.

Figure 3. **Cascade propagation is strongly suppressed in the presence of network isolators.** **a** We consider the six-regular graph shown in Figure 1f with unit edge weights and $7 \cdot 10^4$ different initial conditions where we randomly assign 25% of the nodes to be sources with $P_i = 2$ and the remaining ones to be sinks with $P_i = -\frac{2}{3}$. We then simulate the failure of any possible link in the left module of the network for each initial condition and monitor the size of the resulting cascade of failures, setting the line limit to $F_{i \rightarrow j}^{\max} = 1.0$ (see Methods). We compare two different graphs: the six-regular graph containing a network isolator (light green, dotted) and a corresponding six-regular graph where no links have been rewired (dark green). **b,c** For both graphs, we compare the cascade sizes in the module where the failure was triggered (b) and the other module (c). As a result, cascade sizes are significantly smaller if the other module is shielded by a network isolator although the overall connectivity between the modules is higher in this case. **d-f** We perform the same set of simulations for the graph shown in panel d which confirms the result of reduced cascade sizes in the presence of network isolators. Parameters for panels d-f are given by $P_i = 0.9$ for sources and $P_i = -0.3$ for sinks.

Figure 4. **Robust design of network isolators in weakly non-linear systems.** **a** We analyse a network consisting of two modules that are connected via three links and add a fourth link (dotted) to create a network isolator. We randomly assign 25% of the nodes to be generator nodes (squares) and the remaining ones to be load nodes (triangle). We then simulate the removal of a single link (red) and monitor the corresponding response in the dynamic (b,c) and static (d-f) nonlinear system described by the second order Kuramoto model. **b** We analyse the effect of edge distance (x-axis) and non-linearity on the isolation effect by comparing the flow changes $|\Delta F(\ell)|$ after the failure of a link in the right module (top) and the left, isolated module (bottom) with an increasing degree of non-linearity. As expected, flow changes in the right module are lowest for a weakly non-linear system (bright line) and increase with the non-linearity. However, a strong isolation effect persists even for the highest degree of non-linearity considered here (black lines). **c** To systematically study the isolator’s shielding capabilities, we fix the overall available edge weight of the four edges building the isolator to $\sum_i a_i = 4$ and systematically scan over the remaining degrees of freedom, measuring the performance by means of the mean logarithmic flow changes $\langle \log_{10}(|\Delta F|) \rangle$. **d** We evaluate the available worst-case $N - 1$ weight, i.e. the overall edge weight connecting the two modules after the failure of a single link in the isolator for the same set of edge weights as in c.

Reviewer #2 (Remarks to the Author):

The authors addressed my concerns and questions in an extensive and detailed fashion. In particular I appreciate extending the discussion of non-linear effects and the inclusion of new numerical simulations. At this point I am satisfied with the manuscript and I recommend it for publication in Nature Communications.

Minor comments:

Figure 5 of the main paper: Description of panel B is a little bit difficult to follow. In particular the fact that authors refer to the "top" cluster of curves as ones corresponding to the behavior of the lower module of analyzed network, and the reverse case corresponds to the "bottom" set of curves. Without making a distinction that we talk about those curves, I am starting to look for two separate panels of plots and I get confused. I also do not see the black line that should correspond to the highest considered level of non-linearity. I see a deep purple line, and the only black line on the plot is the arrow pointing up with annotation "non-linearity".

Figure 11 of the Supplement. The labeling in the captions seem to be off. What is labeled as B and C should to me be what is presented on panels C and E, respectively. Then last two lines of captions should refer to panels B, D, F rather than D-F.

Reviewer #4 (Remarks to the Author):

I have reviewed the authors' responses to my comments and the corresponding revisions in the manuscript, and I see that all issues were addressed satisfactorily. I would thus recommend publication of the manuscript after the following additional comments (arising from the revisions) are fully addressed:

- 1) Fig. 4 caption: For clarity, I would mention that the removal of the two nodes in the left gray box in panel c would disconnect the network into two parts and thus form a vertex cut (of size two).
- 2) Fig. 5: I would indicate in the figure, its caption, or in the main text that the increase of nonlinearity corresponds to an increase in Δ or that of $|\tilde{F}|_{\max}$, so that the reader will know what this means without consulting Supplementary Fig. 10 caption. Incidentally, I cannot find the definition of $|\tilde{F}|_{\max}$ in the main text even though it is used in Fig. 5b. This should be defined in the main text.
- 3) Fig. 5: The "available worst-case N-1 weight" appears to measure the damage to the network caused by the failure of the link in the network isolator having the largest weight, but it seems to me that the true cost of such a failure should be measured by a) the maximum flow change and the size of the cascade caused by that failure, and b) the impact on the effectiveness of the network isolator after that failure measured by, e.g., the quantity plotted in Fig. 5c.
- 4) Fig. 5 caption: I would suggest stating that the median absolute flow change is plotted as a function of the distance. Over what was the median taken? From where to where is the distance measured? Please clarify.
- 5) Line 152: Communities/moduli are mentioned here without sufficient explanation. For clarity, I suggest making it clear in the beginning of the section that the network with known communities/moduli is considered.
- 6) Supplementary Fig. 9 caption: "Figure 1f"  "Figure 1c", correct?
- 7) Supplementary Fig. 9 caption: Indicate clearly that the result presented in this figure is for linear flows.

10) Supplementary Fig. 10 caption: "Figure 1f"  "Figure 1c", correct?

11) Supplementary Fig. 12 caption: For panel b (and d and f), it says that the flow is normalized by the "the flow on the initially failing link $F^{\{0\}}_{\text{fail}}$," but if this "initially failing link" refers to the red link in panel a marked by "initial failure" (which is my best guess), the quantity $F^{\{0\}}_{\text{fail}}$ would be a constant, and there is no reason for panels a and b to look different. Can you clarify how the normalization is done?

12) Title: I now realize that the title "Inhibiting failure spreading in complex networks" sounds too general. Since constructing a network isolator to inhibit failure spreading is the main point of the paper, can "network isolator" be incorporated into the title to make it more focused on that message?

Response to Reviewers for:
Inhibiting failure spreading in complex networks

Franz Kaiser,^{1,2} Vito Latora,^{3,4,5,6} and Dirk Witthaut^{1,2}

¹*Forschungszentrum Jülich, Institute for Energy and
Climate Research (IEK-STE), 52428 Jülich, Germany*

²*Institute for Theoretical Physics, University of Cologne, Köln, 50937, Germany*

³*School of Mathematical Sciences, Queen Mary University of London, London E1 4NS, UK*

⁴*Dipartimento di Fisica ed Astronomia,*

Università di Catania and INFN, 95123 Catania, Italy

⁵*The Alan Turing Institute, The British Library, London NW1 2DB, UK*

⁶*Complexity Science Hub Vienna, 1080 Vienna, Austria*

REPLY TO REVIEWER 2

"The authors addressed my concerns and questions in an extensive and detailed fashion. In particular I appreciate extending the discussion of non-linear effects and the inclusion of new numerical simulations. At this point I am satisfied with the manuscript and I recommend it for publication in Nature Communications."

→ We are grateful for this positive feedback and would like to thank the reviewer once again for taking the time to review our paper in detail.

Minor comments:

"Figure 5 of the main paper: Description of panel B is a little bit difficult to follow. In particular the fact that authors refer to the "top" cluster of curves as ones corresponding to the behavior of the lower module of analyzed network, and the reverse case corresponds to the "bottom" set of curves. Without making a distinction that we talk about those curves, I am starting to look for two separate panels of plots and I get confused. I also do not see the black line that should correspond to the highest considered level of non-linearity. I see a deep purple line, and the only black line on the plot is the arrow pointing up with annotation "non-linearity"."

⇒ We are sorry for the confusing notation used here. Indeed, the line we referred to is in fact dark purple. We clarified this part of the figure caption such that it now reads:

"b We consider the median absolute non-linear flow changes $|\Delta\tilde{F}(\ell)|$ (Eq. (11)) on a link ℓ after the removal of the link shown in (a). We analyse the effect of edge distance to the failing link (x-axis) and increasing degree of non-linearity (colour code from light to dark). We compare the flow changes in the lower module that contains the failing link (curves on the upper left) and the isolated module (curves on the lower right) by averaging the flow changes over all links in the given module at a fixed distance. As expected, flow changes in the upper module are lowest for a weakly non-linear system (bright line) and increase with the non-linearity, but a strong isolation effect persists even for a high degree of non-linearity (dark purple line). Shaded region indicates the 0.25- and 0.75-quantiles evaluated over the given distance."

"Figure 11 of the Supplement. The labeling in the captions seem to be off. What is labeled as B and C should to me be what is presented on panels C and E, respectively. Then last two lines of captions should refer to panels B, D, F rather than D-F."

⇒ We thank the reviewer for making us aware of this labeling error and corrected it.

REPLY TO REVIEWER 4

"I have reviewed the authors' responses to my comments and the corresponding revisions in the manuscript, and I see that all issues were addressed satisfactorily. I would thus recommend publication of the manuscript after the following additional comments (arising from the revisions) are fully addressed:"

⇒ We thank the reviewer for this positive assessment of our work and the revision. We would also like to thank the reviewer for taking the time to review our paper once more and for the helpful comments which we address below.

"1) Fig. 4 caption: For clarity, I would mention that the removal of the two nodes in the left gray box in panel c would disconnect the network into two parts and thus form a vertex cut (of size two)."

⇒ We added the following statement to the caption as suggested by the reviewer:

"Note that the removal of these two nodes would disconnect the network into two separate parts, i.e. they form a vertex cut of size two."

"2) Fig. 5: I would indicate in the figure, its caption, or in the main text that the increase of nonlinearity corresponds to an increase in δ or that of $|\tilde{F}|_{\max}$, so that the reader will know what this means without consulting Supplementary Fig. 10 caption. Incidentally, I cannot find the definition of $|\tilde{F}|_{\max}$ in the main text even though it is used in Fig. 5b. This should be defined in the main text."

⇒ We thank the reviewer for this comment and agree that it should be clarified in the main text. We therefore added the following statement to the text:

"To systematically evaluate the degree of non-linearity, we analyse the maximal absolute non-linear flow $|\tilde{F}|_{\max}$ in the entire network. Due to the sinusoidal character of the coupling (see Eq. (11)) and since edge weights are set to unity for the Figure, a relative loading close to unity indicates a highly non-linear system."

"3) Fig. 5: The "available worst-case N-1 weight" appears to measure the damage to the network caused by the failure of the link in the network isolator having the largest weight, but it seems to me that the true cost of such a failure should be measured by a) the maximum flow change and the size of the cascade caused by that failure, and b) the impact on the effectiveness of the network isolator after that failure measured by, e.g., the quantity plotted in Fig. 5c."

⇒ We thank the reviewer for this interesting suggestion. Indeed, the "available worst-case N-1 weight" measures to what extent flow between the two parts of the network is possible after the failure of the link with the largest weight in the isolator. However, we agree that there are other interesting options to measure the cost of a potential failures of this link. In the revised version of the manuscript, we now comment on other possible choices in the main text as follows:

"Note that other choices to estimate the impact of removing a single link in the network isolator, e.g. the size of the cascade caused by the failure of the link in the isolator or the reduction in shielding provided by the isolator after the failure might come to a different conclusion which choice of weights yields the "best" network isolator. "

"4) Fig. 5 caption: I would suggest stating that the median absolute flow change is plotted as a function of the distance. Over what was the median taken? Fro where to where is the distance measured? Please clarify."

⇒ We clarified this part of the figure caption such that it now reads:

"b We consider the median absolute non-linear flow changes $|\Delta\tilde{F}(\ell)|$ (Eq. (11)) on a link ℓ after the removal of the link shown in (a). We analyse the effect of edge distance to the failing link (x-axis) and increasing degree of non-linearity (colour code from light to dark). We compare the flow changes in the lower module that contains the failing link (curves on the upper left) and the isolated module (curves on the lower right) by averaging the flow changes over all links in the given module at a fixed distance. As expected, flow changes in the upper module are lowest for a weakly non-linear system (bright line) and increase with the non-linearity, but a strong isolation effect persists even for a high degree of non-linearity (dark purple line). Shaded region indicates the 0.25- and 0.75-quantiles evaluated over the

given distance.”

”5) Line 152: Communities/moduli are mentioned here without sufficient explanation. For clarity, I suggest making it clear in the beginning of the section that the network with known communities/moduli is considered.”

⇒ We added the following statement to the revised version of the manuscript:

”Here, we assume that the moduli or communities are known for the network under consideration and thus do not address the question how to determine them. ”

”6) Supplementary Fig. 9 caption: ”Figure 1f” -> ”Figure 1c”, correct?”

⇒ We corrected the wrong panel reference as suggested by the reviewer.

”7) Supplementary Fig. 9 caption: Indicate clearly that the result presented in this figure is for linear flows.”

⇒ We updated the corresponding Figure caption by adding a reference to the linear flow model in use here such that it now reads:

”We then simulate the failure of any possible link in the left module of the network for each initial condition using the linear flow model and monitor the size of the resulting cascade of failures, setting the line limit to $F_{i \rightarrow j}^{\max} = 1.0$ (see Methods)”

”10) Supplementary Fig. 10 caption: ”Figure 1f” -> ”Figure 1c”, correct?”

⇒ We corrected the wrong panel reference as suggested by the reviewer.

”11) Supplementary Fig. 12 caption: For panel b (and d and f), it says that the flow is normalized by the ”the flow on the initially failing link $F_{fail}^{(0)}$,” but if this ”initially failing link” refers to the red link in panel a marked by ”initial failure” (which is my best guess), the quantity $F_{fail}^{(0)}$ would be a constant, and there is no reason for panels a and b to look different. Can you clarify how the normalization is done?”

⇒ We are sorry for the confusion. Flow changes are always normalised with respect to the flow carried by the link that fails in the indicated step. Since flow is merely redistributed in the network, this corresponds to a maximum relative flow change of unity in the network (if a single link gets to carry all the flow from the failing link after the failure). We clarified the caption such that it now reads:

”Relative flow changes $|\Delta F_\ell/F_{\text{fail}}^{(0)}|$ on any link ℓ as a result of the failure of the link shown in (a). Flow changes are normalized by the flow carried by the failing link $F_{\text{fail}}^{(0)}$ (a, arrow) before the failure, such that the maximum relative flow change is unity. The flow changes clearly decay with distance from the failing link. **c** Line loading after the initial failure: although flow changes decay with distance, the next failing link is relatively far apart from the initially failing link when considering the geographic or geodesic network distance. **d** Relative flow changes after the failure of the link shown in c, normalised again by the flow that the link shown in c carries before the failure. Again, the flow changes are localised. ”

”12) Title: I now realize that the title ”Inhibiting failure spreading in complex networks” sounds too general. Since constructing a network isolator to inhibit failure spreading is the main point of the paper, can ”network isolator” be incorporated into the title to make it more focused on that message?”

⇒ We agree with the reviewer that the title should be adjusted to incorporate ”network isolators” and thus changed it to ”Network isolators inhibit failure spreading in complex networks”.